# A unifying framework for generalised Bayesian online learning in non-stationary environments

**Gerardo Duran-Martin**                                          *g.duran@me.com*
*School of Mathematical Sciences*
*Queen Mary University of London, UK*

**Leandro Sánchez-Betancourt**                        *sanchezbetan@maths.ox.ac.uk*
*Mathematical Institute &*
*Oxford-Man Institute of Quantitative Finance*
*University of Oxford, UK*

**Alexander Y. Shestopaloff**                              *a.shestopaloff@qmul.ac.uk*
*School of Mathematical Sciences*
*Queen Mary University of London, UK &*
*Department of Mathematics and Statistics*
*Memorial University of Newfoundland, Canada*

**Kevin Murphy**                                          *kpmurphy@google.com*
*Google DeepMind*
*Moutain View, CA, USA*

**Reviewed on OpenReview:** *https://openreview.net/forum?id=osesw2V10u*

## Abstract

We propose a unifying framework for methods that perform probabilistic online learning in non-stationary environments. We call the framework BONE, which stands for generalised (B)ayesian (O)nline learning in (N)on-stationary (E)nvironments. BONE provides a common structure to tackle a variety of problems, including online continual learning, prequential forecasting, and contextual bandits. The framework requires specifying three modelling choices: (i) a model for measurements (e.g., a neural network), (ii) an auxiliary process to model non-stationarity (e.g., the time since the last changepoint), and (iii) a conditional prior over model parameters (e.g., a multivariate Gaussian). The framework also requires two algorithmic choices, which we use to carry out approximate inference under this framework: (i) an algorithm to estimate beliefs (posterior distribution) about the model parameters given the auxiliary variable, and (ii) an algorithm to estimate beliefs about the auxiliary variable. We show how the modularity of our framework allows for many existing methods to be reinterpreted as instances of BONE, and it allows us to propose new methods. We compare experimentally existing methods with our proposed new method on several datasets, providing insights into the situations that make each method more suitable for a specific task. We provide a Jax open source library to facilitate the adoption of this framework.

## 1 Introduction

In this paper, we unify adaptive probabilistic methods that learn to make predictions about the next output $\boldsymbol{y}_{t+1}$ based on the next input $\boldsymbol{x}_{t+1}$ and a sequence of past inputs and outputs, $(\boldsymbol{x}_{1:t}, \boldsymbol{y}_{1:t})$, where $t$ indexes time. This is often called prequential forecasting (Gama et al., 2008), online learning (Zhang, 2023, Chapter 1), or sequential online prediction (Liu, 2023).

Many probabilistic prediction methods assume that the data generating process (DGP) $p(\boldsymbol{y}_{t+1} \,|\, \boldsymbol{x}_{t+1}, \boldsymbol{x}_{1:t}, \boldsymbol{y}_{1:t})$ is static through time. However, real-world data often comes from non-stationary distributions, where the underlying data distribution changes, either gradually (e.g., rising mean temperature) or abruptly (e.g., price shocks after a major news event). In this paper we propose a unified framework for tackling (one-step-ahead) forecasts in such (potentially) non-stationary environments. Building on recent advances in generalised (pseudo) Bayesian methods (Bissiri et al., 2016; Knoblauch et al., 2022; Khan & Rue, 2023), we show that our framework unifies classical Bayesian approaches and naturally extends to methods derived from Bayes' rule. We demonstrate that all these methods are related to a three-layered hierarchical state-space model, where the top layer captures changes in the data, the second layer governs the evolution of model parameters, and the third layer represents the observations.

Our framework accommodates and extends many existing lines of work, including online continual learning (Dohare et al., 2024), prequential (one-step-ahead) forecasting (Liu, 2023), test-time adaptation (Schirmer et al., 2024), neural contextual bandits (Riquelme et al., 2018), filtering (Basseville et al., 1993), and changepoint detection or segmentation (Gupta et al., 2024). For example, methods that were originally developed to tackle changepoint detection can be reinterpreted and applied within our framework to address a wider range of sequential online learning problems. This flexibility allows us to gather tools that were designed for specific tasks—often studied in isolation within distinct subfields—into a unifying framework for handling non-stationarity across various application domains.

The framework we propose in this paper, which we call BONE—which stands for (B)ayesian (O)nline learning in (N)on-stationary (E)nvironments—is based on a form of generalised Bayesian inference in a hierarchical model, and is composed of three modelling choices and two algorithmic choices. The modelling choices are: (M.1) a model for the measurements, (M.2) an auxiliary process to model non-stationarity, and (M.3) a prior over model parameters conditioned on the auxiliary process and the past data. The algorithmic choices are: (A.1) an algorithm to compute an approximate posterior over the model parameters given the auxiliary variables, and (A.2) an algorithm to compute an approximate posterior—or more generally, a set of weights—over the auxiliary variables. We show how these different axes of variation span a wide variety of existing and new methods.[1] To illustrate this, Figure 1 shows how various methods can be written within the BONE framework. Here, we categorised methods by how the auxiliary variable is used to model non-stationarity and how it influences the prior over model parameters before observing a new data point.

Examples demonstrating the use of an easy-to-use library that implements these methods, written in Jax (Bradbury et al., 2018), are available at `https://github.com/gerdm/BONE`.

To summarise, our contributions are threefold: (1) we provide an extensive literature review on methods that tackle non-stationarity, and show that they can all be written as instances of our unified BONE framework; (2) we use the BONE framework to develop a new method; and (3) we perform an experimental comparison of many existing methods and the new method on environments with both abrupt changes and gradually-changing distributions.

## 2 The framework

Consider a sequence of measurements $\boldsymbol{y}_{1:T} = (\boldsymbol{y}_1, \ldots, \boldsymbol{y}_T)$ with $\boldsymbol{y}_i \in \mathbb{R}^d$, and (unmodelled) features $\boldsymbol{x}_{1:T} = (\boldsymbol{x}_1, \ldots, \boldsymbol{x}_T)$ with $\boldsymbol{x}_i \in \mathbb{R}^q$. Let $\mathcal{D}_t = (\boldsymbol{x}_t, \boldsymbol{y}_t)$ be a datapoint and $\mathcal{D}_{1:T} = (\mathcal{D}_1, \ldots, \mathcal{D}_T)$ the dataset at time $T$. Consider a conditional probabilistic model for $\boldsymbol{y}_t$, conditioned on $\boldsymbol{x}_t$, and parametrised by $\boldsymbol{\theta}_t \in \mathbb{R}^m$, given by $p(\boldsymbol{y}_t \,|\, \boldsymbol{\theta}_t, \boldsymbol{x}_t)$, and sometimes called the likelihood.

Assume that the conditional mean for $\boldsymbol{y}_t$ is encoded in a parametric model $h(\boldsymbol{\theta}_t, \boldsymbol{x}_t)$, e.g., a neural network. A popular modelling choice for regression problems is to assume that $p(\boldsymbol{y}_t \,|\, \boldsymbol{\theta}_t, \boldsymbol{x}_t) = \mathcal{N}(\boldsymbol{y}_t \,|\, h(\boldsymbol{\theta}_t, \boldsymbol{x}_t), \mathbf{R}_t)$ for known covariance $\mathbf{R}_t$. Next, assume there exists an auxiliary variable $\psi_t \in \boldsymbol{\Psi}_t$ that evolves following the

---

[1]BONE is designed for sequential prediction with time-indexed data, thus, it does not aim to find a "global" fit over all possible data points. Instead, BONE focuses on adaptive strategies that forecast the next datapoint (one at a time). This is in contrast to global optimisation approaches, such as Luo et al. (2024), which focus on finding the best global model for all datapoints. There are a number of probabilistic methods that handle non-stationarity and are not included in BONE. For example, Scalzo et al. (2021) and de Vilmarest & Wintenberger (2021) do not decouple the modelling of non-stationarity and the modelling over model parameters, thus, they cannot be written within our formulation.

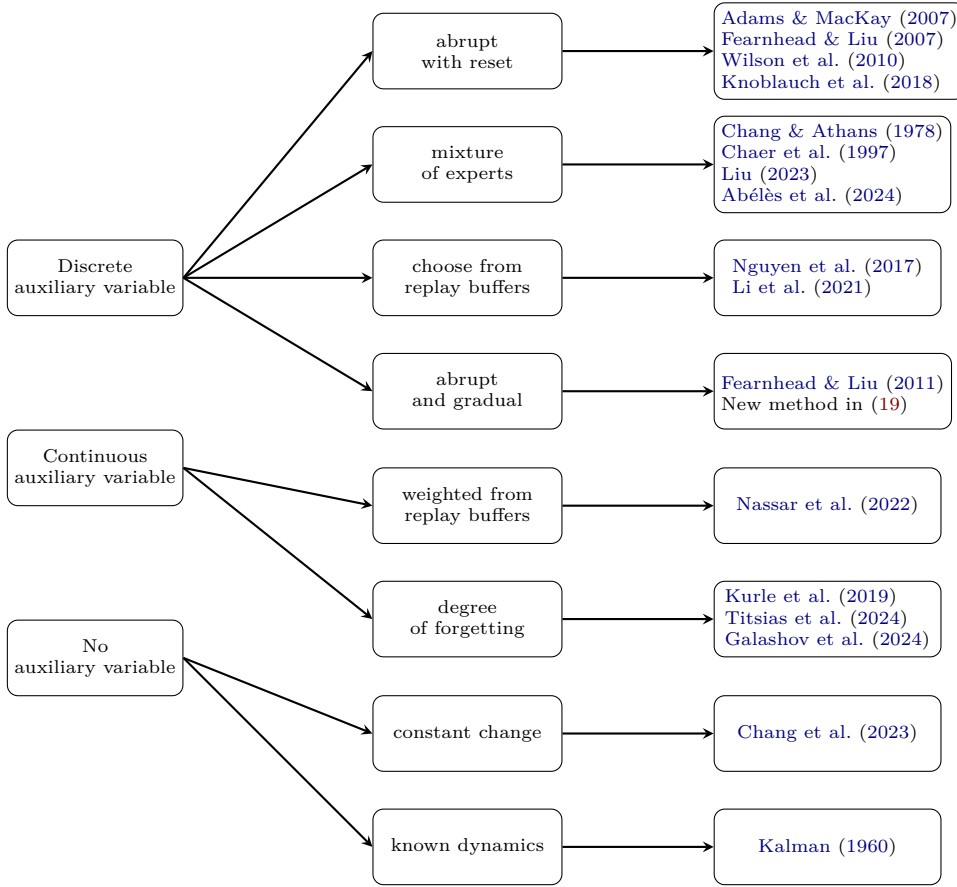

Figure 1: Overview of BONE methods grouped by the qualitative nature of the auxiliary variable and the conditional prior. See Table 3 for a detailed breakdown of these methods.

dynamics $p(\psi_t \,|\, \psi_{t-1})$ and encodes information about the non-stationarity of the sequence at time $t$. Here, $\boldsymbol{\Psi}_t$ is the set of possible values of the auxiliary variable $\psi_t$. The purpose of this variable is, for instance, to determine which past datapoints $\boldsymbol{y}_{1:t-1}$ most closely align with the most recent measurement $\boldsymbol{y}_t$. We describe the auxiliary variable in detail in Section 2.4. Finally, the model parameters $\boldsymbol{\theta}_t$ evolve following the dynamics $p(\boldsymbol{\theta}_t \,|\, \boldsymbol{\theta}_{t-1}, \psi_t)$. This represents how much parameters change, given the state of the auxiliary variable.

Figure 2 shows the probabilistic graphical model that motivates our formulation; this resembles the one in Doucet et al. (2000) with an additional optional dependence between the auxiliary variable and the measurements; in what follows we omit this dependence for brevity.

For an experiment of length $T \in \mathbb{N}$, the joint conditional density over the model parameters, induced by the graphical model shown in Figure 2, is given by

$$p(\boldsymbol{y}_{1:T}, \boldsymbol{\theta}_{0:T}, \psi_{0:T} \,|\, \boldsymbol{x}_{1:T}) = p(\boldsymbol{\theta}_0)\, p(\psi_0) \prod_{t=1}^{T} p(\boldsymbol{y}_t \,|\, \boldsymbol{\theta}_t, \boldsymbol{x}_t)\, p(\boldsymbol{\theta}_t \,|\, \boldsymbol{\theta}_{t-1}, \psi_t)\, p(\psi_t \,|\, \psi_{t-1}). \qquad (1)$$

In this paper, we are interested in methods that efficiently compute the so-called *expected* posterior predictive $\hat{\boldsymbol{y}}_{t+1} := \mathbb{E}_{p(\boldsymbol{\theta}_t, \psi_t \,|\, \mathcal{D}_{1:t})}[h(\boldsymbol{\theta}_t, \boldsymbol{x}_{t+1})]$ in an online and recursive manner. In our setting, one observes $\boldsymbol{x}_{t+1}$ just before observing $\boldsymbol{y}_{t+1}$; thus, to make a prediction about $\boldsymbol{y}_{t+1}$, we have $\boldsymbol{x}_{t+1}$ and $\mathcal{D}_{1:t}$ at our disposal.[2] For the case of a discrete auxiliary variable $\psi_t \in \Psi_t$, the form of the expected posterior predictive for $\boldsymbol{y}_{t+1}$, induced

---

[2]The input features $\boldsymbol{x}_{t+1}$ and output measurements $\boldsymbol{y}_{t+1}$ can correspond to different time steps. For example, $\boldsymbol{x}_{t+1}$ can be the state of the stock market at a fixed date and $\boldsymbol{y}_{t+1}$ is the return on a stock some days into the future.

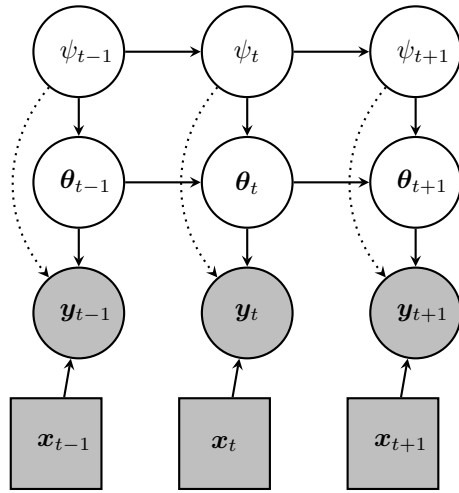

Figure 2: Two-levelled hierarchical state-space model (SSM) with known dynamics, motivating our BONE framework Solid arrows indicate required dependencies, while dashed arrows represent optional dependencies. Rectangles denote exogenous variables, and circles represent random variables. Observed elements are shaded in gray. The left shift in $\boldsymbol{x}_t$ represents that features are observed before observing $\boldsymbol{y}_t$.

by (1), is

$$
\begin{aligned}
\hat{\boldsymbol{y}}_{t+1} &= \mathbb{E}_{p(\boldsymbol{\theta}_t, \psi_t \mid \mathcal{D}_{1:t})}[h(\boldsymbol{\theta}_t, \boldsymbol{x}_{t+1})] \\
&= \sum_{\psi_t \in \Psi_t} \int h(\boldsymbol{\theta}_t, \boldsymbol{x}_{t+1})\, p(\boldsymbol{\theta}_t, \psi_t \mid \mathcal{D}_{1:t}) \mathrm{d}\boldsymbol{\theta}_t \\
&= \sum_{\psi_t \in \Psi_t} p(\psi_t \mid \mathcal{D}_{1:t}) \int h(\boldsymbol{\theta}_t, \boldsymbol{x}_{t+1}) p(\boldsymbol{\theta}_t \mid \psi_t, \mathcal{D}_{1:t}) \mathrm{d}\boldsymbol{\theta}_t,
\end{aligned}
\tag{2}
$$

where

$$
p(\boldsymbol{\theta}_t \mid \psi_t, \mathcal{D}_{1:t}) \propto p(\boldsymbol{y}_t \mid \boldsymbol{\theta}_t, \boldsymbol{x}_t)\, p(\boldsymbol{\theta}_t \mid \psi_t, \mathcal{D}_{1:t-1}),
\tag{3}
$$

$$
p(\boldsymbol{\theta}_t \mid \psi_t, \mathcal{D}_{1:t-1}) = \int p(\boldsymbol{\theta}_t \mid \boldsymbol{\theta}_{t-1}, \psi_t)\, p(\boldsymbol{\theta}_{t-1} \mid \mathcal{D}_{1:t-1})\, \mathrm{d}\boldsymbol{\theta}_{t-1},
\tag{4}
$$

$$
p(\psi_t \mid \mathcal{D}_{1:t}) = p(\boldsymbol{y}_t \mid \boldsymbol{x}_t, \psi_t, \mathcal{D}_{1:t-1}) \sum_{\psi_{t-1} \in \boldsymbol{\Psi}_{t-1}} p(\psi_{t-1} \mid \mathcal{D}_{1:t-1})\, p(\psi_t \mid \psi_{t-1}, \mathcal{D}_{1:t-1}).
\tag{5}
$$

From (2), (3), (4), and (5) we argue that there are three key modelling choices and two algorithmic choices. Specifically, the three key modelling choices are: (M.1) the conditional mean $h(\boldsymbol{\theta}, \boldsymbol{x})$ together with the likelihood $p(\boldsymbol{y} \mid \boldsymbol{\theta}, \boldsymbol{x})$; (M.2) the auxiliary variable $\psi_t$; and (M.3) the conditional prior $p(\boldsymbol{\theta}_t \mid \psi_t, \mathcal{D}_{1:t-1})$. Additionally, the two algorithmic choices are: (A.1) the algorithm to compute (or approximate) the conditional posterior over model parameters $p(\boldsymbol{\theta}_t \mid \psi_t, \mathcal{D}_{1:t})$, and (A.2) the algorithm that computes (or approximates) the posterior over weights $p(\psi_t, \mid \mathcal{D}_{1:t})$.

The BONE framework generalises these choices, allowing for greater flexibility while maintaining the motivating probabilistic structure. Instead of the likelihood model $p(\boldsymbol{y}_t \mid \boldsymbol{\theta}_t, \boldsymbol{x}_t)$ with conditional mean $h(\boldsymbol{\theta}_t, \boldsymbol{x}_t)$, we consider a general function $\exp(-\ell(\boldsymbol{y}_t; \boldsymbol{\theta}_t, \boldsymbol{x}_t))$, where $\ell(\boldsymbol{y}_t; \boldsymbol{\theta}_t, \boldsymbol{x}_t)$ could be either a loss function or a log-likelihood. Next, instead of the conditional prior $p(\boldsymbol{\theta}_t \mid \psi_t, \mathcal{D}_{1:t-1})$, we introduce a more general modelling function, $\pi(\boldsymbol{\theta}_t; \psi_t, \mathcal{D}_{1:t-1})$ that governs the prior over model parameters.[3] Similarly, instead of the posterior density $p(\boldsymbol{\theta}_t \mid \psi_t, \mathcal{D}_{1:t})$, we employ the function $q(\boldsymbol{\theta}_t; \psi_t, \mathcal{D}_{1:t})$; e.g., an approximation of the posterior, or

---

[3]This function adopts an ad hoc approach to parameter evolution instead of explicitly solving the integration step (4). In Appendix C, we detail how the inference is carried out when the dynamics are known. More precisely, we conduct a full-Bayesian treatment by calculating the posterior of model parameters at $t+1$ and using this for inference.

a generalised posterior (Bissiri et al., 2016). Finally, instead of the posterior over weights $p(\psi_t \,|\, \mathcal{D}_{1:t})$, we consider a weighting function $\nu(\psi_t; \mathcal{D}_{1:t})$, which can be the Bayesian posterior or an ad-hoc time-dependent weighting function.

This generalisation is important because it unifies a wide range of existing methods under a common framework. Many well-known approaches in the literature can be written as elements of BONE by appropriately selecting the model for measurements, the conditional prior, and the posterior approximations. BONE highlights connections between different methods, it also enables systematic comparisons under a common umbrella, and it allows us to develop novel algorithms. Table 1 explicitly contrasts the choices in BONE with those in the classical Bayesian formalism.

| component | BONE | Bayes |
|---|---|---|
| (M.1: model) | $h(\boldsymbol{\theta}_t, \boldsymbol{x}_t)$ & $\exp(-\ell(\boldsymbol{y}_t; \boldsymbol{\theta}_t, \boldsymbol{x}_t))$ | $h(\boldsymbol{\theta}_t, \boldsymbol{x}_t)$ & $p(\boldsymbol{y}_t \,|\, \boldsymbol{\theta}_t, \boldsymbol{x}_t)$ |
| (M.2: auxvar) | $\psi_t$ | $\psi_t$ |
| (M.3: prior) | $\pi_t(\boldsymbol{\theta}_t; \psi_t) := \pi(\boldsymbol{\theta}_t; \psi_t, \mathcal{D}_{1:t-1})$ | $p(\boldsymbol{\theta}_t \,|\, \psi_t, \mathcal{D}_{1:t-1})$ |
| (A.1: posterior) | $q_t(\boldsymbol{\theta}_t; \psi_t) := q(\boldsymbol{\theta}_t; \psi_t, \mathcal{D}_{1:t})$ | $p(\boldsymbol{\theta}_t \,|\, \psi_t, \mathcal{D}_{1:t})$ |
| (A.2: weighting) | $\nu_t(\psi_t) := \nu(\psi_t; \mathcal{D}_{1:t})$ | $p(\psi_t \,|\, \mathcal{D}_{1:t})$ |

Table 1: Components of the BONE framework.

With these modifications, the expected posterior predictive under BONE is

$$\hat{\boldsymbol{y}}_t := \sum_{\psi_t \in \Psi_t} \underbrace{\nu(\psi_t \,|\, \mathcal{D}_{1:t})}_{\text{(A.2: weighting)}} \int \underbrace{h(\boldsymbol{\theta}_t, \boldsymbol{x}_{t+1})}_{\text{(M.1: model)}} \underbrace{q(\boldsymbol{\theta}_t; \psi_t, \mathcal{D}_{1:t})}_{\text{(A.1: posterior)}} \mathrm{d}\boldsymbol{\theta}_t, \tag{6}$$

where

$$q(\boldsymbol{\theta}_t; \psi_t, \mathcal{D}_{1:t}) \propto \underbrace{\pi(\boldsymbol{\theta}_t; \psi_t, \mathcal{D}_{1:t-1})}_{\text{(M.3: prior)}} \underbrace{\exp(-\ell(\boldsymbol{y}_t; \boldsymbol{\theta}_t, \boldsymbol{x}_t))}_{\text{(M.1: model)}} \tag{7}$$

takes the form of a generalised posterior (Bissiri et al., 2016). In classical Bayesian setting, the loss function takes the form of the negative log-likelihood, i.e.,

$$\ell(\boldsymbol{y}_t; \boldsymbol{\theta}_t, \boldsymbol{x}_t) = -\log p(\boldsymbol{y}_t \,|\, \boldsymbol{\theta}_t, \boldsymbol{x}_t). \tag{8}$$

Unless stated otherwise, we work with the negative log-likelihood in (8) found in the classical Bayesian setting.

A prediction for $\boldsymbol{y}_{t+1}$ given $\mathcal{D}_{1:t}$, $\boldsymbol{x}_{t+1}$, and $\psi_t$ is

$$\hat{\boldsymbol{y}}_{t+1}^{(\psi_t)} = \mathbb{E}_{q_t}[h(\boldsymbol{\theta}_t; \boldsymbol{x}_{t+1}) \,|\, \psi_t] := \int h(\boldsymbol{\theta}_t; \boldsymbol{x}_{t+1})\, q(\boldsymbol{\theta}_t; \psi_t, \mathcal{D}_{1:t})\mathrm{d}\boldsymbol{\theta}_t. \tag{9}$$

Here we use the shorthand notation $q_t = q(\boldsymbol{\theta}_t; \psi_t, \mathcal{D}_{1:t})$ and $\mathbb{E}_{q_t}[\cdot \,|\, \psi_t]$ to highlight dependence on $\psi_t$.

Algorithm 1 provides pseudocode for the prediction and update steps in the BONE framework. Notably, these components can be broadly divided into two categories: modelling and algorithmic. The modelling components determine the inductive biases in the model, and correspond to $h$, $\ell$, $\psi_t$, and $\pi$. The algorithmic components dictate how operations are carried out to produce a final prediction — this corresponds to $q_t$ and $\nu_t$.

---

**Algorithm 1** Generic predict and update step of BONE with discrete $\psi_t$ at time $t$.

---

**Require:** $\mathcal{D}_{1:t}$ // past data
**Require:** $\boldsymbol{x}_{t+1}$ // optional inputs
**Require:** $h(\boldsymbol{\theta}, \boldsymbol{x}_t)$ // Choice of (M.1: model)
**Require:** $\boldsymbol{\Psi}_t$ // Choice of (M.2: auxvar)
 1: **for** $\psi_t \in \boldsymbol{\Psi}_t$ **do**
 2:    $\pi_t(\boldsymbol{\theta}_t; \psi_t) \leftarrow \pi(\boldsymbol{\theta}_t; \psi_t, \mathcal{D}_{1:t-1})$ // choice of (M.3: prior)
 3:    $q_t(\boldsymbol{\theta}_t; \psi_t) \leftarrow q(\boldsymbol{\theta}_t; \psi_t, \mathcal{D}_{1:t}) \propto \pi_t(\boldsymbol{\theta}_t \psi_t) \exp(-\ell(\boldsymbol{y}_t; \boldsymbol{\theta}_t, \boldsymbol{x}_t))$// choice of (A.1: posterior)
 4:    $\nu_t(\psi_t) \leftarrow \nu(\psi_t; \mathcal{D}_{1:t})$ // choice of (A.2: weighting)
 5:    $\hat{\boldsymbol{y}}_{t+1}^{(\psi_t)} \leftarrow \mathbb{E}_{q_t}[h(\boldsymbol{\theta}_t, \boldsymbol{x}_{t+1}); \psi_t]$ // conditional prequential prediction
 6: **end for**
 7: $\hat{\boldsymbol{y}}_{t+1} \leftarrow \sum_{\psi_t} \nu_t(\psi_t) \, \hat{\boldsymbol{y}}_{t+1}^{(\psi_t)}$ // weighted prequential prediction

---

## 2.1 Example tasks that can be solved with BONE

Before going into more detail about BONE, we give some concrete examples of tasks in which the BONE framework can be applied. We group these examples into unsupervised tasks and supervised tasks.

### 2.1.1 Unsupervised tasks

Unsupervised tasks involve estimating unobservable quantities of interest from the data $\mathcal{D}_{1:t}$. Below, we present three common tasks in this category.

**Segmentation**    Segmentation involves partitioning the data stream into contiguous subsequences or "blocks" $\{\mathcal{D}_{1:t_1}, \mathcal{D}_{t_1+1:t_2}, \ldots\}$, where the DGP for each block is governed by a sequence of unknown functions (Barry & Hartigan, 1992). The goal is to determine the points in time when a new block begins, known as changepoints. This is useful in many applications, such as finance, where detecting changes in market trends is critical. In this setting, non-stationarity is assumed to be abrupt and occurring at unknown points in time. We study an example in Section 4.3.1. For a survey of segmentation methods, see e.g, Aminikhanghahi & Cook (2017); Gupta et al. (2024).

**Filtering using state-space models (SSM)**    Filtering estimates an underlying latent state $\boldsymbol{\theta}_t$ that evolves over time (often representing a meaningful concept). The posterior estimate of $\boldsymbol{\theta}_t$ is computed by applying Bayesian inference to the corresponding state space model (SSM), which determines the choice of (M.1: model), and how the state changes over time, through the choice of (M.3: prior). Examples include estimating the state of the atmosphere (Evensen, 1994), tracking the position of a moving object (Battin, 1982), or recovering a signal from a noisy system (Basseville et al., 1993). In this setting, non-stationarity is usually assumed to be continuous and occurring at possible time-varying rates. For a survey of filtering methods, see e.g., Chen et al. (2003).

**Segmentation using Switching state-space models (SSSM)**    In this task, the modeller extends the standard SSM with a set of discrete latent variables $\psi_t \in \{1, \ldots, K\}$, which may change value at each time step according to a state transition matrix. The parameters of the rest of the DGP depend on the discrete state $\psi_t$. The objective is to infer the sequence of underlying discrete states that best "explains" the observed data (Ostendorf et al., 1996; Ghahramani & Hinton, 2000; Beal et al., 2001; Fox et al., 2007; Van Gael et al., 2008; Linderman et al., 2017). In this context, non-stationarity arises from the switching behaviour of the underlying discrete process.

### 2.1.2 Supervised tasks

Supervised tasks involve predicting a measurable outcome $\boldsymbol{y}_t$. Unlike unsupervised tasks, this allows the performance of the model to be assessed objectively, since we can compare the prediction to the actual observation. We present three common tasks in this category below.

**Prequential forecasting**  Prequential (or one-step-ahead) forecasting (Gama et al., 2008) seeks to predict the value $\boldsymbol{y}_{t+1}$ given $\mathcal{D}_{1:t}$ and $\boldsymbol{x}_{t+1}$. This is distinct from time-series forecasting, which typically does not consider exogenous variables $\boldsymbol{x}_t$, and thus can forecast (or "roll out") many steps into the future. We study an example in Section 4.1. For a survey on prequential forecasting under non-stationarity, see e.g., Lu et al. (2018).

**Online continual learning (OCL)**  OCL is a broad term used for learning regression or classification models online, typically with neural networks. These methods usually assume that the underlying data generating mechanism could shift. The objective of OCL methods is to train a model that performs consistently across both past and future data, rather than just focusing on future forecasting (Cai et al., 2021). The changepoints (corresponding to different "tasks") may or may not be known. This setting addresses the stability-plasticity dilemma, focusing on retaining previously learned knowledge while adapting to new tasks. We study an example of OCL for classification, when the task boundaries are not known, in Section 4.1.2. For a survey on recent methods for OCL, see e.g., Gunasekara et al. (2023).

**(Non-stationary) contextual bandits**  In contextual bandit problems, the agent is presented with features $\boldsymbol{x}_{t+1}$, and must choose an action (arm) that yields the highest expected reward (Li et al., 2010). We let $\boldsymbol{y}_{t+1} \in \mathbb{R}^A$ where $A > 2$ is the number of possible actions; this is a vector where the $a$-th entry contains the reward one would have obtained had one chosen arm $a$. Let $\boldsymbol{y}_t^{(a)}$ be the observed reward at time $t$ after choosing arm $a$, i.e., the $a$-th entry of $\boldsymbol{y}_t$. A popular approach for choosing the optimal action (while tackling the exploration-exploitation tradeoff) at each step is Thomson sampling (TS) (Thompson, 1933), which in our setting works as follows: first, sample a parameter vector from the posterior, $\tilde{\boldsymbol{\theta}}_t$ from $q(\boldsymbol{\theta}_t; \psi_t, \mathcal{D}_{1:t})$; then, greedily choose the best arm (the one with the highest expected payoff), $a_{t+1} = \arg\max_a \hat{\boldsymbol{y}}_{t+1}^{(a)}$, where $\hat{\boldsymbol{y}}_{t+1} = h(\tilde{\boldsymbol{\theta}}_t; \boldsymbol{x}_{t+1})$; and $\hat{\boldsymbol{y}}_{t+1}^{(a)}$ is the $a$-th entry of $\hat{\boldsymbol{y}}_{t+1}$; finally, receive a reward $\boldsymbol{y}_{t+1}^{(a_{t+1})}$. The goal is to select a sequence of arms $\{a_1, \ldots, a_T\}$ that maximises the cumulative reward $\sum_{t=1}^{T} \boldsymbol{y}_t^{(a_t)}$. When the mapping function $h$ is a neural network, this model is called a neural bandit. TS for neural bandits has been studied in many papers, see e.g., Duran-Martin et al. (2022) and references therein. Non-stationary bandits have been studied in Mellor & Shapiro (2013); Cartea et al. (2023a); Alami (2023); Liu et al. (2023). We study an example in Section 4.2.

## 2.2  Details of BONE

In the following subsections, we describe each component of the BONE framework in detail, provide illustrative examples, and reference relevant literature for further reading.

## 2.3  The measurement model (M.1)

Recall that $h(\boldsymbol{\theta}, \boldsymbol{x})$ is a parametric model that encodes the conditional mean for $\boldsymbol{y}$, given $\boldsymbol{\theta}$ and $\boldsymbol{x}$. For linear measurement models, $h(\boldsymbol{\theta}, \boldsymbol{x})$ is given by:

$$h(\boldsymbol{\theta}, \boldsymbol{x}) = \begin{cases} \boldsymbol{\theta}^{\mathsf{T}}\boldsymbol{x} & \text{(regression)}, y \in \mathbb{R} \\ \sigma(\boldsymbol{\theta}^{\mathsf{T}}\boldsymbol{x}) & \text{(binary classification)}, y \in \{0, 1\} \\ \text{Softmax}(\boldsymbol{\theta}^{\mathsf{T}}\boldsymbol{x}) & \text{(multi-class classification)}, \boldsymbol{y} \in \{0, 1\}^C \end{cases} \tag{10}$$

where $\sigma(z) = (1 + \exp(-z))^{-1}$ is the sigmoid function, $C \in \mathbb{N}$ is the number of classes, $\text{Softmax}(\boldsymbol{z})_k = \exp(\boldsymbol{z}_k)/\sum_i \exp(\boldsymbol{z}_i)$ represents the softmax function with $\boldsymbol{z} \in \mathbb{R}^d$ and $\boldsymbol{z}_i$ the $i$-th element of $\boldsymbol{z}$. In the machine learning literature, the vector $\boldsymbol{z}$ is called the logits of the classifier. For non-linear measurement models, such as neural networks, $h(\boldsymbol{\theta}, \boldsymbol{x})$ represents the output of the network parameterised by $\boldsymbol{\theta}$. The best choice of $h$ will depend on the nature of the data, as well as the nature of the task, in particular, whether it is supervised or unsupervised. We give some examples in Section 4.

### 2.4 The auxiliary variable (M.2)

The choice of auxiliary variable $\psi_t$ is crucial to identify changes in the data-generating process, allowing our framework to track non-stationarity. Below, we give a list of the common auxiliary variables used in the literature.

RL (runlength): $\psi_t = r_t \in \{0, \ldots, t\}$ is a scalar representing a *lookback window*, defined as the number of steps since the last regime change. The value $r_t = 0$ indicates the start of a new regime at time $t$, while $r_t \geq 1$ denotes the continuation of a regime with a lookback window of length $r_t$. This choice of auxiliary variable is common in the changepoint detection literature. See e.g., Adams & MacKay (2007); Knoblauch et al. (2018); Alami et al. (2020); Agudelo-España et al. (2020); Altamirano et al. (2023); Alami (2023). This auxiliary variable is useful for non-stationary data with non-repeating temporal segments, provided we know the intensity with which new segments appear.

RLCC (runlength and changepoint count): $\psi_t = (r_t, c_t) \in \{0, \ldots, t\} \times \{0, \ldots, t\}$ is a vector that represents both the runlength and the total number of changepoints, as proposed in Wilson et al. (2010). When $r_t = t$, this implies $c_t = 0$, meaning no changepoints have occurred. Conversely, $r_t = 0$ indicates the start of a new regime and implies $c_t \in \{1, \ldots, t\}$, accounting for at least one changepoint. For a given $r_t \geq 0$, the changepoint count $c_t$ belongs to the range $\{1, \ldots, t - r_t\}$. As with RL, this auxiliary variable assumes consecutive time blocks, but additionally allows us to estimate the likelihood of entering a new regime by tracking the number of changepoints seen so far. This auxiliary variable is useful for non-stationary data with non-repeating temporal segments when the intensity with which new segments appear is unknown.

CPT (changepoint timestep): $\psi_t = \zeta_t$, with $\zeta_t = \{\zeta_{1,t}, \ldots, \zeta_{\ell,t}\}$, is a set of size $\ell \in \{0, \ldots t\}$ containing the $\ell$ times at which there was a changepoint, with the convention that $0 \leq \zeta_{1,t} < \zeta_{2,t} < \ldots < \zeta_{\ell,t} \leq t$. This choice of auxiliary variable was introduced in Fearnhead & Liu (2007) and has been studied in Fearnhead & Liu (2011); Fearnhead & Rigaill (2019). Under mild assumptions, it can be shown that CPT is equivalent to RL, see e.g., Knoblauch & Damoulas (2018). This auxiliary variable is useful for non-stationary data with non-repeating temporal segments when the probability of a new segment appearing is unknown and knowledge of the changepoint location is required.

CPL (changepoint location): $\psi_t = s_{1:t} \in \{0,1\}^t$ is a binary vector. In one interpretation, $s_i = 1$ indicates the occurrence of a changepoint at time $i$, as in Li et al. (2021), while in another, it means that $\mathcal{D}_t$ belongs to the current regime, as in Nassar et al. (2022). This auxiliary variable is useful for non-stationary data with repeating temporal segments. It is useful when the segments are formed of non-consecutive datapoints.

CPV (changepoint probability vector): $\psi_t = v_{1:t} \in (0,1)^t$ is a $t$-dimensional random vector representing the probability of each element in the history belonging to the current regime. This generalises CPL and was introduced in Nassar et al. (2022) for online continual learning, allowing for a more fine-grained representation of changepoints over time. This auxiliary variable is useful for non-stationary data with repeating temporal segments. Unlike CPL, it takes a vector of weights in $(0, 1)$ which allows for higher flexibilty when compared to CPL.

CPP (changepoint probability): $\psi_t = v_t \in (0,1)$ represents the probability of a changepoint. This is a special case of CPV that tracks only the most recent changepoint probability; this choice was used in Titsias et al. (2024) for online continual learning.

ME (mixture of experts): $\psi_t = \alpha_t \in \{1, \ldots, K\}$ represents one of $K$ experts. Each expert corresponds to either a choice of model or one of $K$ possible hyperparameters. This approach has been applied to filtering (Chaer et al., 1997) and prequential forecasting (Liu, 2023; Abélès et al., 2024). This auxiliary variable facilitates the weighting of predictions made by models when one has a fixed number of competing models.

C: $\psi_t = c$ represents a constant auxiliary variable, where $c$ is just a placeholder or dummy value. This is equivalent to not having an auxiliary variable, or alternatively, to having a single expert that encodes all available information.

**Space-time complexity** There is a tradeoff between the complexity that $\psi$ is able to encode and the computation power needed to perform updates. Loosely speaking, this can be seen in the cardinality of

the set of possible values of $\psi$ through time. Let $\boldsymbol{\Psi}_t$ be the space of possible values for $\psi_t$. Depending on the choice of $\psi_t$, the cardinality of $\boldsymbol{\Psi}_t$ either stay constant or increase over time, i.e., $\boldsymbol{\Psi}_{t-1} \subseteq \boldsymbol{\Psi}_t$ for all $t = 1, \ldots, T$. For instance, the possible values for RL increase by one at each timestep; the possible values of CPL double at each timestep; finally, the possible values for ME do not increase. Table 2 shows the space of values and cardinality that $\boldsymbol{\Psi}_t$ takes as a function of the choice of auxiliary variable.

| name | C | CPT | CPP | CPL | CPV | ME | RL | RLCC |
|---|---|---|---|---|---|---|---|---|
| values | $\{c\}$ | $2^{\{0,1,\ldots,t\}}$ | $[0,1]$ | $\{0,1\}^t$ | $(0,1)^t$ | $\{1,\ldots,K\}$ | $\{0,1,\ldots,t\}$ | $\{\{0,t\},\ldots,\{t,0\}\}$ |
| cardinality | 1 | $2^t$ | $\infty$ | $2^t$ | inf | $K$ | $t$ | $2 + t(t+1)/2$ |

Table 2: Design space for the auxiliary random variables $\psi_t$. Here, $T$ denotes the total number of timesteps and $K$ denotes a fixed number of candidates.

## 2.5 Conditional prior (M.3)

This component defines the prior predictive distribution over model parameters conditioned on the choice of (M.2: auxvar) $\psi_t$ and the dataset $\mathcal{D}_{1:t-1}$. In some cases, explicit access to past data is not needed.

For example, a common assumption is to have a Gaussian conditional prior over model parameters. In this case, we assume that, given data $\mathcal{D}_{1:t-1}$ and the auxiliary variable $\psi_t$, the conditional prior takes the form

$$\pi(\boldsymbol{\theta}_t; \psi_t, \mathcal{D}_{1:t-1}) = \mathcal{N}\big(\boldsymbol{\theta}_t \,|\, g_t(\psi_t, \mathcal{D}_{1:t-1}), G_t(\psi_t, \mathcal{D}_{1:t-1})\big), \tag{11}$$

with $g_t : \boldsymbol{\Psi}_t \times \mathbb{R}^{(m+d)(t-1)} \to \mathbb{R}^m$ a function that returns the mean vector of model parameters, $\mathbb{E}[\boldsymbol{\theta}_t \,|\, \psi_t, \mathcal{D}_{1:t-1}]$, and $G_t : \boldsymbol{\Psi}_t \times \mathbb{R}^{(m+d)(t-1)} \to \mathbb{R}^{m\times m}$ a function that returns a $m$-dimensional covariance matrix, $\mathrm{Cov}[\boldsymbol{\theta}_t \,|\, \psi_t, \mathcal{D}_{1:t-1}]$. In what follows, we let $(\boldsymbol{\mu}_0, \boldsymbol{\Sigma}_0)$ be the pre-defined initial prior mean and covariance. Furthermore, we denote $(\boldsymbol{\mu}_{t-1}, \boldsymbol{\Sigma}_{t-1})$ be the posterior mean and covariance found at time $t-1$, which is used as a prior at time $t$.

Below, we provide a non-exhaustive list of possible combinations of choices for (M.2: auxvar) and (M.3: prior) of the form (11) that can be found in the literature, and we also introduce a new combination.

C-LSSM (constant linear with affine state-space model). We assume the parameter dynamics can be modeled by a linear-Gaussian state space model (LSSM), i.e., $\mathbb{E}[\boldsymbol{\theta}_t \,|\, \boldsymbol{\theta}_{t-1}] = \mathbf{F}_t \boldsymbol{\theta}_{t-1} + \boldsymbol{b}_t$ and $\mathrm{Cov}[\boldsymbol{\theta}_t \,|\, \boldsymbol{\theta}_{t-1}] = \mathbf{Q}_t$, for given $(m \times m)$ dynamics matrix $\mathbf{F}_t$, $(m \times 1)$ bias vector $\boldsymbol{b}_t$, and $(m \times m)$ positive semi-definite matrix $\mathbf{Q}_t$. We also assume $\psi_t = c$ is a fixed (dummy) constant, which is equivalent to not having an auxiliary variable. The characterisation of the conditional prior takes the form

$$\begin{aligned} g_t(c, \mathcal{D}_{1:t-1}) &= \mathbf{F}_t \boldsymbol{\mu}_{t-1} + \boldsymbol{b}_t, \\ G_t(c, \mathcal{D}_{1:t-1}) &= \mathbf{F}_t \boldsymbol{\Sigma}_{t-1} \mathbf{F}_t^\intercal + \mathbf{Q}_t , \end{aligned} \tag{12}$$

This is a common baseline model that we will specialise below.

C-OU (constant with Ornstein-Uhlenbeck process). This is a special case of the C-LSSM model where $\mathbf{F}_t = \gamma \mathbf{I}$, $\boldsymbol{b}_t = (1-\gamma)\boldsymbol{\mu}_0$, $\mathbf{Q}_t = (1-\gamma^2)\boldsymbol{\Sigma}_0$, $\boldsymbol{\Sigma}_0 = \sigma_0^2 \mathbf{I}$, $\gamma \in [0,1]$ is the fixed rate, and $\sigma_0 \geq 0$. The conditional prior mean and covariance are a convex combination of the form

$$\begin{aligned} g(c, \mathcal{D}_{1:t-1}) &= \gamma \boldsymbol{\mu}_{t-1} + (1-\gamma)\boldsymbol{\mu}_0, \\ G(c, \mathcal{D}_{1:t-1}) &= \gamma^2 \boldsymbol{\Sigma}_{t-1} + (1-\gamma^2)\boldsymbol{\Sigma}_0. \end{aligned} \tag{13}$$

This combination is used in Kurle et al. (2019). Smaller values of the rate parameter $\gamma$ correspond to a faster resetting, i.e., the distribution of model parameters revert more quickly to the prior belief $(\boldsymbol{\mu}_0, \boldsymbol{\Sigma}_0)$, which means the past data will be forgotten.

CPP-OU (changepoint probability with Ornstein-Uhlenbeck process). Here $\psi_t = \upsilon_t \in [0,1]$ is the changepoint probability that we use as the rate of an Ornstein-Uhlenbeck (OU) process, as proposed in Titsias et al. (2024); Galashov et al. (2024). The characterisation of the conditional prior takes the form

$$\begin{aligned} g(\upsilon_t, \mathcal{D}_{1:t-1}) &= \upsilon_t \boldsymbol{\mu}_{t-1} + (1-\upsilon_t)\boldsymbol{\mu}_0 , \\ G(\upsilon_t, \mathcal{D}_{1:t-1}) &= \upsilon_t^2 \boldsymbol{\Sigma}_{t-1} + (1-\upsilon_t^2)\boldsymbol{\Sigma}_0 . \end{aligned} \tag{14}$$

An example on how to compute $v_t$ using an empirical Bayes procedure is given in (42).

**C-ACI** (constant with additive covariance inflation). This corresponds to a special case of **C-LSSM** in which $\mathbf{F} = \mathbf{I}$, $\boldsymbol{b} = \mathbf{0}$, and $\mathbf{Q} = \alpha\mathbf{I}$ for $\alpha > 0$ is the amount of noise added at each step. This combination is used in Kuhl (1990); Duran-Martin et al. (2022); Chang et al. (2022; 2023) . The characterisation of the conditional prior takes the form

$$
\begin{aligned}
g(c, \mathcal{D}_{1:t-1}) &= \boldsymbol{\mu}_{t-1}, \\
G(c, \mathcal{D}_{1:t-1}) &= \boldsymbol{\Sigma}_{t-1} + \mathbf{Q}_t.
\end{aligned}
\tag{15}
$$

This is similar to **C-OU** with $\gamma = 1$, however, here we inject new noise at each step. Another variant of this scheme, known as *shrink-and-perturb* (Ash & Adams, 2020), takes $g(c, \mathcal{D}_{1:t-1}) = q\,\boldsymbol{\mu}_{t-1}$ and $G(c, \mathcal{D}_{1:t-1}) = \boldsymbol{\Sigma}_{t-1} + \mathbf{Q}_t$, where $0 < q < 1$ is the shrinkage parameters, and $\mathbf{Q}_t = \sigma_0^2\,\mathbf{I}$.

**C-Static** (constant with static parameters). Here $\psi_t = c$ (with $c$ a dummy variable). This is a special case of the **C-ACI** configuration in which $\alpha = 0$. The conditional prior is characterised by

$$
\begin{aligned}
g_t(c, \mathcal{D}_{1:t-1}) &= \boldsymbol{\mu}_{t-1}, \\
G_t(c, \mathcal{D}_{1:t-1}) &= \boldsymbol{\Sigma}_{t-1}.
\end{aligned}
\tag{16}
$$

**ME-LSSM** (mixture of experts with LSSM). Here $\psi_t = \alpha_t \in \{1, \ldots, K\}$, and we have a bank of $K$ independent LSSM models; the auxiliary variable specifies which model to use at each step. The characterisation of the conditional prior takes the form

$$
\begin{aligned}
g_t(\alpha_t, \mathcal{D}_{1:t-1}) &= \mathbf{F}_t^{(\alpha_t)}\,\boldsymbol{\mu}_{t-1}^{(\alpha_t)} + \boldsymbol{b}_t^{(\alpha_t)}, \\
G_t(\alpha_t, \mathcal{D}_{1:t-1}) &= \mathbf{F}_t^{(\alpha_t)}\,\boldsymbol{\Sigma}_{t-1}^{(\alpha_t)}\mathbf{F}_t^{\intercal} + \mathbf{Q}_t^{(\alpha_t)}.
\end{aligned}
\tag{17}
$$

The superscript $(\alpha_t)$ denotes the conditional prior for the $k$-th expert. More precisely, $\boldsymbol{\mu}_{t-1}^{(\alpha_t)}, \boldsymbol{\Sigma}_{t-1}^{(\alpha_t)}$ are the posterior at time $t-1$ using $\mathbf{F}_{t-1}^{(\alpha_t)}$ and $\mathbf{Q}_{t-1}^{(\alpha_t)}$ from the $k$-th expert. This combination was introduced in Chaer et al. (1997).

**RL-PR** (runlength with prior reset): for $\psi_t = r_t$, this choice of auxiliary variable constructs a new mean and covariance considering the past $t - r_t$ observations. We have

$$
\begin{aligned}
g_t(r_t, \mathcal{D}_{1:t-1}) &= \boldsymbol{\mu}_0\,\mathbb{1}(r_t = 0) + \boldsymbol{\mu}_{(r_{t-1})}\mathbb{1}(r_t > 0), \\
G_t(r_t, \mathcal{D}_{1:t-1}) &= \boldsymbol{\Sigma}_0\,\mathbb{1}(r_t = 0) + \boldsymbol{\Sigma}_{(r_{t-1})}\mathbb{1}(r_t > 0),
\end{aligned}
\tag{18}
$$

where $\boldsymbol{\mu}_{(r_{t-1})}, \boldsymbol{\Sigma}_{(r_{t-1})}$ denotes the posterior belief computed using observations from indices $t - r_t$ to $t - 1$. The case $r_t = 0$ corresponds to choosing the initial pre-defined prior mean and covariance $\boldsymbol{\mu}_0$ and $\boldsymbol{\Sigma}_0$. This combination assumes that data from a single regime arrives in sequential *blocks* of time of length $r_t$. This choice of (M.3: prior) was first studied in Adams & MacKay (2007).

**RL[1]-OUPR\*** (greedy runlength with OU and prior reset): This is a new combination we consider in this paper, which is designed to accommodate both gradual changes and sudden changes. More precisely, we assume $\psi_t = r_t$, and we choose the conditional prior as either a hard reset to the prior, if $v_t(r_t) > \varepsilon$, or a convex combination of the prior and the previous belief state (using an **OU** process), if $v_t(r_t) \leq \varepsilon$. That is, we define the conditional prior as

$$
g_t(r_t, \mathcal{D}_{1:t-1}) = \begin{cases} \boldsymbol{\mu}_0\,(1 - v_t(r_t)) + \boldsymbol{\mu}_{(r_t)}\,v_t(r_t) & v_t(r_t) > \varepsilon, \\ \boldsymbol{\mu}_0 & v_t(r_t) \leq \varepsilon, \end{cases}
\tag{19}
$$

$$
G_t(r_t, \mathcal{D}_{1:t-1}) = \begin{cases} \boldsymbol{\Sigma}_0\,(1 - v_t(r_t)^2) + \boldsymbol{\Sigma}_{(r_t)}\,v_t(r_t)^2 & v_t(r_t) > \varepsilon, \\ \boldsymbol{\Sigma}_0 & v_t(r_t) \leq \varepsilon. \end{cases}
\tag{20}
$$

Here $v_t(r_t) = p(r_t \mid \mathcal{D}_{1:t})$, with $r_t = r_{t-1} + 1$, is the probability we are continuing a segment, and $v_t(r_t)$ with $r_t = 0$ is the probability of a changepoint. For details on how to compute $v_t(r_t)$, see (37). The value of the threshold parameter $\varepsilon$ controls whether an abrupt change or a gradual change should take place. In the limit

when $\varepsilon = 1$, this new combination does not learn, since it is always doing a hard reset to the initial beliefs at time $t = 0$. Conversely, when $\varepsilon = 0$, we obtain an OU-type update weighted by $\nu_t$. When $\varepsilon = 0.5$, we revert back to prior beliefs when the most likely hypothesis is that a changepoint has just occurred. Finally, we remark that the above combination allows us to make use of non-Markovian choices for (M.1: model), as we see in Section 4.3.1. This is, to the best of our knowledge, a new combination that has not been proposed in the previous literature; for further details see Appendix A.3.

CPL-Sub (changepoint location with subset of data): for $\psi_t = s_{1:t}$, this conditional prior constructs the mean and covariance as

$$
\begin{aligned}
g_t(s_{1:t}, \mathcal{D}_{1:t-1}) &= \boldsymbol{\mu}_{(s_{1:t-1})}, \\
G_t(s_{1:t}, \mathcal{D}_{1:t-1}) &= \boldsymbol{\Sigma}_{(s_{1:t-1})},
\end{aligned}
\tag{21}
$$

where $\boldsymbol{\mu}_{(s_{1:t-1})}, \boldsymbol{\Sigma}_{(s_{1:t-1})}$ denotes the posterior belief computed using the observations for entries where $s_{1:t-1}$ have value of 1. This combination assumes that data from the current regime is scattered from the past history. That is, it assumes that data from a past regime could become relevant again at a later date. This combination has been studied in Nguyen et al. (2017).

CPL-MCI (changepoint location with multiplicative covariance matrix): for $\psi_t = s_{1:t}$, this choice of conditional prior maintains the prior mean, but increases the norm of the prior covariance by a constant term $\beta \in (0, 1)$. More precisely, we have that

$$
\begin{aligned}
g_t(s_{1:t}, \mathcal{D}_{1:t-1}) &= \boldsymbol{\mu}_{(s_{1:t-1})}, \\
G_t(s_{1:t}, \mathcal{D}_{1:t-1}) &= \begin{cases} \beta^{-1} \boldsymbol{\Sigma}_{(s_{1:t-1})} & s_t = 1, \\ \boldsymbol{\Sigma}_{(s_{1:t-1})} & s_t = 0. \end{cases}
\end{aligned}
\tag{22}
$$

This combination was first proposed in Li et al. (2021).

CPT-MMPR (changepoint timestep using moment-matched prior reset): for $\psi_t = \boldsymbol{\zeta}_t$, with $\boldsymbol{\zeta}_t = \{\zeta_{1,t}, \ldots, \zeta_{\ell,t}\}$, and $\zeta_{\ell,t}$ the position of the last changepoint, the work of Fearnhead & Liu (2011) assumes a dependence structure between changepoints. That is, to build the conditional prior mean and covariance, they consider the past $\mathcal{D}_{\zeta_{\ell,t}:t-1}$ datapoints whenever $\zeta_{\ell,t} \leq t - 1$ and a moment-matched approximation to the mixture density over all possible subset densities since the last changepoint whenever $\zeta_{\ell,t} = t$. For an example of MMPR with RL choice of (M.2: auxvar), see Appendix A.2.

## 2.6 Algorithm to compute the posterior over model parameters (A.1)

This section presents algorithms for estimating the density $q(\boldsymbol{\theta}_t; \psi_t, \mathcal{D}_{1:t})$; we focus on methods that yield Gaussian posterior densities. Specifically, we are interested in practical approaches for approximating the conditional Bayesian posterior, as given in (7).

There is a vast body of literature on methods for estimating the posterior over model parameters. Here, we focus on three common approaches for computing the Gaussian posterior: conjugate updates (Cj), linear-Gaussian approximations (LG), and variational Bayes (VB). For an overview of choices of (A.1: posterior) and a comparison in terms of their computational complexity, see Table 3 in Jones et al. (2024).

### 2.6.1 Conjugate updates (Cj)

Conjugate updates (Cj) provide a classical approach for computing the posterior by leveraging conjugate prior distributions. Conjugate updates occur when the functional form of the conditional prior $\pi(\boldsymbol{\theta}_t; \psi_t, \mathcal{D}_{1:t-1})$ matches that of the measurement model $p(\boldsymbol{y}_t \mid \boldsymbol{\theta}, \boldsymbol{x}_t)$ (Robert et al., 2007, Section 3.3). This property allows the posterior to remain within the same family as the prior, which leads to analytically tractable updates and facilitates efficient recursive estimation.

A common example is the conjugate Gaussian model, where the measurement model is Gaussian with known variance and the prior is a multivariate Gaussian. This results in closed-form updates for both the mean and covariance. Another example is the Beta-Bernoulli pair, where the measurement model follows a Bernoulli

distribution with an unknown probability, and the prior is a Beta distribution. See e.g., Bernardo & Smith (1994); West & Harrison (1997) for details.

The recursive nature of conjugate updates makes them particularly useful for real-time or sequential learning scenarios, where fast and efficient updates are crucial.

### 2.6.2 Linear-Gaussian approximation (`LG`)

The linear-Gaussian (`LG`) method builds on the conjugate updates (`Cj`) above. More precisely, the prior is Gaussian and the measurement model is approximated by a linear Gaussian model, which simplifies computations.

The prior over model parameters is taken as:

$$\pi(\boldsymbol{\theta}_t; \psi_t, \mathcal{D}_{1:t-1}) = \mathcal{N}\left(\boldsymbol{\theta}_t \mid \boldsymbol{\mu}_{t-1}^{(\psi_t)}, \boldsymbol{\Sigma}_{t-1}^{(\psi_t)}\right), \tag{23}$$

where $\boldsymbol{\mu}_{t-1}^{(\psi_t)}$ and $\boldsymbol{\Sigma}_{t-1}^{(\psi_t)}$ are the mean and covariance, respectively. We use the measurement function $h$ to define a first-order approximation $\bar{h}_t$ around the prior mean which is given by

$$\bar{h}_t(\boldsymbol{\theta}_t, \boldsymbol{x}_t) = h\left(\boldsymbol{\mu}_{t-1}^{(\psi_t)}, \boldsymbol{x}_t\right) + \mathbf{H}_t\left(\boldsymbol{\theta}_t - \boldsymbol{\mu}_{t-1}^{(\psi_t)}\right). \tag{24}$$

Here, $\mathbf{H}_t$ is the Jacobian of $h(\boldsymbol{\theta}, \boldsymbol{x}_t)$ with respect to $\boldsymbol{\theta}$, evaluated at $\boldsymbol{\mu}_{t-1}^{(\psi_t)}$. The approximate posterior measure is given by

$$\begin{aligned}
q(\boldsymbol{\theta}_t; \psi_t, \mathcal{D}_{1:t}) &\propto \mathcal{N}(\boldsymbol{y}_t \mid \bar{h}_t(\boldsymbol{\theta}_t, \boldsymbol{x}_t), \mathbf{R}_t)\, \pi(\boldsymbol{\theta}_t; \psi_t, \mathcal{D}_{1:t-1}) \\
&= \mathcal{N}(\boldsymbol{y}_t \mid \bar{h}_t(\boldsymbol{\theta}_t, \boldsymbol{x}_t), \mathbf{R}_t)\, \mathcal{N}\left(\boldsymbol{\theta}_t \mid \boldsymbol{\mu}_{t-1}^{(\psi_t)}, \boldsymbol{\Sigma}_{t-1}^{(\psi_t)}\right) \\
&\propto \mathcal{N}(\boldsymbol{\theta}_t \mid \boldsymbol{\mu}_t^{(\psi_t)}, \boldsymbol{\Sigma}_t^{(\psi_t)}),
\end{aligned} \tag{25}$$

where $\mathbf{R}_t$ is a known noise covariance matrix of the measurement $\boldsymbol{y}_t$. Under the `LG` algorithmic choice, the updated equations are

$$\begin{aligned}
\hat{\boldsymbol{y}}_t^{(\psi_t)} &= h\left(\boldsymbol{\mu}_{t-1}^{(\psi_t)}, \boldsymbol{x}_t\right), \\
\mathbf{S}_t^{(\psi_t)} &= \mathbf{H}_t\, \boldsymbol{\Sigma}_{t-1}^{(\psi_t)}\, \mathbf{H}_t^\mathsf{T} + \mathbf{R}_t, \\
\mathbf{K}_t^{(\psi_t)} &= \boldsymbol{\Sigma}_{t-1}^{(\psi_t)}\, \mathbf{H}_t^\mathsf{T}\left(\mathbf{S}_t^{(\psi_t)}\right)^{-1}, \\
\boldsymbol{\mu}_t^{(\psi_t)} &= \boldsymbol{\mu}_{t-1}^{(\psi_t)} + \mathbf{K}_t^{(\psi_t)}\left(\boldsymbol{y}_t - \hat{\boldsymbol{y}}_t^{(\psi_t)}\right), \\
\boldsymbol{\Sigma}_t^{(\psi_t)} &= \boldsymbol{\Sigma}_{t-1}^{(\psi_t)} - \left(\mathbf{K}_t^{(\psi_t)}\right)\left(\mathbf{S}_t^{(\psi_t)}\right)\left(\mathbf{K}_t^{(\psi_t)}\right)^\mathsf{T}.
\end{aligned} \tag{26}$$

This linear approximation enables efficient computation of the posterior in a Gaussian form. Examples include the extended Kalman filter (EKF) (Haykin, 2004), which applies local linearisation to non-linear systems, the exponential family EKF (Ollivier, 2018), which approximates the measurement model as Gaussian by matching the first two moments, and the low-rank Kalman filter (LoFi) method (Chang et al., 2023), which assumes a diagonal-plus-low-rank (DLR) posterior precision matrix. See Särkkä & Svensson (2023) for more details on such Gaussian filtering methods.

### 2.6.3 Variational Bayes (`VB`)

Variational Bayes (VB) is a popular method for approximating a posterior distribution of model parameters by choosing a parametric family (such as Gaussians) that is computationally tractable. The primary objective of VB is to minimise the Kullback-Leibler (KL) divergence between a candidate Gaussian distribution and the density $q_t$. It can be shown that we can safely ignore the normalisation constant for $q_t$, which is often computationally expensive, so we can replace $q_t$ with its unnormalised form. We have the following optimisation problem for the posterior variational parameters:

$$(\boldsymbol{\mu}_t, \boldsymbol{\Sigma}_t) = \underset{\boldsymbol{\mu}, \boldsymbol{\Sigma}}{\arg\min}\, \mathbf{D}_{\mathrm{KL}}\left(\mathcal{N}(\boldsymbol{\theta}_t \mid \boldsymbol{\mu}, \boldsymbol{\Sigma}) \,\|\, p(\boldsymbol{y}_t \mid \boldsymbol{\theta}_t, \boldsymbol{x}_t)\, \pi(\boldsymbol{\theta}_t; \psi_t, \mathcal{D}_{1:t-1})\right), \tag{27}$$

where $\pi_t(\boldsymbol{\theta}_t; \psi_t)$ is the chosen prior distribution (M.3: prior).

An example of VB for neural network models is the Bayes-by-backpropagation method (BBB) of Blundell et al. (2015), which assumes a diagonal posterior covariance (more expressive forms are also possible). Nguyen et al. (2017) extended BBB to non-stationary settings. More recent approaches involve recursive estimation, such as the recursive variational Gaussian approximation (R-VGA) method of Lambert et al. (2022) which uses a full rank Gaussian variational approximation; the low-rank RVGA (L-RVGA) method of Lambert et al. (2023), which uses a diagonal plus low-rank (DLR) Gaussian variational approximation; the Bayesian online natural gradient (BONG) method of Jones et al. (2024), which combines the DLR approximation with EKF-style linearisation for additional speedups; the natural gradient Gaussian approximation (NANO) method of Cao et al. (2024), which uses a diagonal Gaussian approximation similar to VD-EKF in Chang et al. (2022); and the projection-based unification of last-layer and subspace estimation (PULSE) method of Cartea et al. (2023b), which targets different posterior densities for a subspace of the hidden layers and a full-rank covariance over the final layer of a neural network.

### 2.6.4 Alternative methods

Alternative approaches for handling nonlinear or nonconjugate measurements have been proposed, such as sequential Monte Carlo (SMC) methods (de Freitas et al., 2000), and ensemble Kalman filters (EnKF) (Roth et al., 2017). These sample-based methods are particularly advantageous when the dimensionality of $\boldsymbol{\theta}$ is large, or when a more exact posterior approximation is required, providing greater flexibility in non-linear and non-Gaussian environments.

Generalised Bayesian methods, such as Mishkin et al. (2018); Knoblauch et al. (2022), generalise the VB update of (27) by allowing the right-hand side to be a loss function. Alternatively, online gradient descent methods like Bencomo et al. (2023) emulate state-space modelling via gradient-based optimisation.

### 2.7 Weighting function for auxiliary variable (A.2)

The term $\nu_t(\psi_t)$ defines the weights over possible values of the auxiliary variable (M.2: auxvar). We compute it as the marginal posterior distribution $\nu_t(\psi_t) = p(\psi_t \mid \mathcal{D}_{1:t})$ (see e.g., Adams & MacKay (2007); Fearnhead & Liu (2007; 2011); Li et al. (2021)) or with *ad-hoc* rules (see e.g., Nassar et al. (2022); Abélès et al. (2024); Titsias et al. (2024)). In the former case, the weighting function takes the form

$$
\begin{aligned}
\nu_t(\psi_t) &= p(\psi_t \mid \mathcal{D}_{1:t}) \\
&= p(\boldsymbol{y}_t \mid \boldsymbol{x}_t, \psi_t, \mathcal{D}_{1:t-1}) \int_{\psi_{t-1} \in \boldsymbol{\Psi}_{t-1}} p(\psi_{t-1} \mid \mathcal{D}_{1:t-1})\, p(\psi_t \mid \psi_{t-1}, \mathcal{D}_{1:t-1}) \mathrm{d}\psi_{t-1},
\end{aligned}
\tag{28}
$$

where one assumes that $\boldsymbol{y}_t$ is conditionally independent of $\psi_{t-1}$, given $\psi_t$, and $\boldsymbol{x}_t$ is an exogenous vector. The first term on the right hand side of (28) is known as the conditional posterior predictive, and is given by

$$
p(\boldsymbol{y}_t \mid \boldsymbol{x}_t, \psi_t, \mathcal{D}_{1:t-1}) = \int p(\boldsymbol{y}_t \mid \boldsymbol{\theta}_t, \boldsymbol{x}_t)\, \pi(\boldsymbol{\theta}_t; \psi_t, \mathcal{D}_{1:t-1}) \mathrm{d}\boldsymbol{\theta}_t.
\tag{29}
$$

This integral over $\boldsymbol{\theta}_t$ may require approximations, as we discussed in Section 2.6. Furthermore, the integral over $\psi_{t-1}$ in (28) may also require approximations, depending on the nature of the auxiliary variable $\psi_t$, and the modelling assumptions for $p(\psi_t \mid \psi_{t-1}, \mathcal{D}_{1:t-1})$. We provide some examples below.

### 2.7.1 Discrete auxiliary variable (DA)

Here we assume the auxiliary variable takes values in a discrete space $\psi_t \in \boldsymbol{\Psi}_t$. The weights for the discrete auxiliary variable (DA) can be computed with a fixed number of hypotheses $K \geq 1$ or with a growing number of hypotheses if the cardinality of $\boldsymbol{\Psi}_t$ increases through time; we denote these cases by DA[K] and DA[inf] respectively. Below, we provide three examples that estimate the weights under DA[inf] recursively.

RL (runlength with Markovian assumption): for $\psi_t = r_t$, the work of Adams & MacKay (2007) takes

$$p(r_t \mid r_{t-1}, \mathcal{D}_{1:t-1}) = \begin{cases} 1 - H(r_{t-1}) & \text{if } r_t = r_{t-1} + 1, \\ H(r_{t-1}) & \text{if } r_t = 0, \\ 0 & \text{otherwise,} \end{cases} \tag{30}$$

where $H : \mathbb{N}_0 \to (0,1)$ is the hazard function. A popular choice is to take $H(r) = \kappa \in (0,1)$ to be a fixed constant hyperparameter known as the hazard rate. The choice `RL[inf]-PR` is known as the Bayesian online changepoint detection model (BOCD).

CPL (changepoint location): for $\psi_t = s_{1:t}$, the work of Li et al. (2021) takes

$$p(\tilde{s}_{1:t} \mid s_{1:t-1}, \mathcal{D}_{1:t-1}) = \begin{cases} \kappa & \text{if } ([\tilde{s}_{1:t} \setminus \tilde{s}_t] = s_{1:t-1}) \text{ and } \tilde{s}_t = 1, \\ 1 - \kappa & \text{if } ([\tilde{s}_{1:t} \setminus \tilde{s}_t] = s_{1:t-1}) \text{ and } \tilde{s}_t = 0, \\ 0 & \text{otherwise,} \end{cases} \tag{31}$$

i.e., the sequence of changepoints at time $t$ correspond to the sequence of changepoints up to time $t-1$, plus a newly sampled value for $t$. See Appendix A.4 for details on how to compute $\nu_t(s_{1:t})$.

CPT (changepoint timestep with Markovian assumption): for $\psi_t = \boldsymbol{\zeta}_t$, the work of Fearnhead & Liu (2007) takes

$$p(\boldsymbol{\zeta}_t \mid \boldsymbol{\zeta}_{t-1}, \mathcal{D}_{1:t-1}) = p(\zeta_{\ell,t} \mid \zeta_{\ell,t-1}) = J(\zeta_{\ell,t} - \zeta_{\ell,t-1}), \tag{32}$$

with $J : \mathbb{N}_0 \to (0,1)$ a probability mass function. Note that $\zeta_{\ell,t} - \zeta_{\ell,t-1}$ is the distance between two changepoints, i.e., a runlength. In this sense, $\zeta_{\ell,t} - \zeta_{\ell,t-1} = r_t$, which relates `CPT` to `RL`. See their paper for details on how to compute $\nu_t(\boldsymbol{\zeta}_t)$.

**Low-memory variants — from `DA[inf]` to `DA[K]`** In the examples above, the number of computations to obtain $\sum_{\psi_t} \nu_t(\psi_t)$ grows in time. To fix the computational cost, one can restrict the sum to be over a subset $\mathcal{A}_t$ of the space of $\psi_t$ with cardinality $|\mathcal{A}_t| = K \geq 1$. Each element in the set $\mathcal{A}_t$ is called a hypothesis and given $K \geq 1$, we keep the $K$ most likely elements —according to $\nu_t(\psi_t)$— in $\mathcal{A}_t$. We then define the normalised weighting function

$$\hat{\nu}_t(\psi_t) = \frac{\nu_t(\psi_t)}{\sum_{\psi_t' \in \mathcal{A}_t} \nu_t(\psi_t')}, \tag{33}$$

which we use instead of $\nu_t(\psi_t)$. For example, in `RL` above, $\mathcal{A}_{t-1} = \{r_{t-1}^{(k)} : k = 1, \ldots, K\}$ are the unique $K$ most likely runlengths where the superscript represents the ranking according to $\nu_{t-1}(\cdot)$. Then, at time $t$, the augmented set $\bar{\mathcal{A}}_t$ becomes $(\mathcal{A}_{t-1} + 1) \cup \{0\}$, where the sum is element-wise, and we then compute the $K$ most likely elements of $\bar{\mathcal{A}}_t$ to define $\mathcal{A}_t$. In `CPL`, $\mathcal{A}_{t-1}$ contains the $K$ most likely sequences of changepoints, $\bar{\mathcal{A}}_t$ is defined as the collection of the $2K$ sequences where each sequence of $\mathcal{A}_{t-1}$ has a zero or one concatenated at the end. Finally, the $K$ most likely elements in $\bar{\mathcal{A}}_t$ define $\mathcal{A}_t$. This style of pruning is common in segmentation methods; see, e.g., Saatçi et al. (2010), but other pruning are also possible, such as those proposed by Li et al. (2021), or sampling-based approaches; see e.g. Doucet et al. (2000).

**Other choices for `DA[K]`** Finally, some choices of weighting functions are derived using ad-hoc rules, meaning that explicit or approximate solutions to the Bayesian posterior are not needed. One of the most popular choices of ad-hoc weighting functions are mixture of experts, which weight different models according to a given criterion.

ME (mixture of experts with algorithmic weighting): Consider $\psi_t = \boldsymbol{\alpha}_t$. Let $\boldsymbol{\alpha}_{t,k} = k$ denote the $k$-th *configuration* over (M.3: prior). Next, denote by $\boldsymbol{w}_t = \{\boldsymbol{w}_{t,1}, \ldots, \boldsymbol{w}_{t,K}\}$ a set of weights, where $\boldsymbol{w}_{t,k}$ corresponds to the weight for the $k$-th expert at time $t$. The work of Chaer et al. (1997) considers the weighting function

$$\nu_t(\boldsymbol{w}_t)_k = \frac{\exp(\boldsymbol{w}_{t,k}^{\mathsf{T}} \boldsymbol{y}_t)}{\sum_{j=1}^{K} \exp(\boldsymbol{w}_{t,j}^{\mathsf{T}} \boldsymbol{y}_t)}, \tag{34}$$

for $k = 1, \ldots, K$. The set of weights $\boldsymbol{w}_t$ are determined by maximising the surrogate gain function

$$\mathcal{G}_t(\boldsymbol{w}_t) = p(\boldsymbol{y}_t \mid \boldsymbol{x}_t, \mathcal{D}_{1:t-1}) = \sum_{k=1}^{K} p(\boldsymbol{y}_t \mid \boldsymbol{x}_t, \boldsymbol{\alpha}_{t,k}, \mathcal{D}_{1:t-1}) \, \nu_t(\boldsymbol{w}_t)_k, \tag{35}$$

with respect to $\boldsymbol{w}_{t,k}$ for all $k = 1, \ldots, K$ at every timestep $t$.

We write `DA[K]`, where `K` is the number hypothesis, for methods that use `K` hypotheses at most. On the other hand, we write `DA[inf]` when we do not impose a bound on the number of hypotheses used. Note that even when the choice of (A.2: weighting) is built using `DA[inf]`, one can modify it to make it `DA[K]`.

**Discrete auxiliary variable with greedy hypothesis selection (`DA[1]`)** A special case of the above is `DA[1]`, where we employ a single hypothesis. In these scenarios, we set $\nu(\psi_t) = 1$ where $\psi_t$ is the most likely hypothesis.

RL (Greedy runlength): For $\psi_t = r_t$ and `DA[1]`, we take

$$p(r_t \mid r_{t-1}, \mathcal{D}_{1:t-1}) = \begin{cases} 1 - \kappa & \text{if } r_t = r_{t-1} + 1, \\ \kappa & \text{if } r_t = 0, \\ 0 & \text{otherwise.} \end{cases} \tag{36}$$

Our choice of (A.2: weighting) is based on the marginal predictive likelihood ratio, which is derived from the computation of $p(r_t \mid \mathcal{D}_{1:t})$ under either either an increase in the runlength ($r_t^{(1)} = r_{t-1} + 1$) or a reset of the runlength ($r_t^{(0)} = 0$). Under these assumptions, the form of $\nu_t(r_t^{(1)})$ is

$$\nu_t(r_t^{(1)}) = \frac{p(\boldsymbol{y}_t \mid r_t^{(1)}, \boldsymbol{x}_t, \mathcal{D}_{1:t-1}) \, (1 - \kappa)}{p(\boldsymbol{y}_t \mid r_t^{(0)}, \boldsymbol{x}_t, \mathcal{D}_{1:t-1}) \, \kappa + p(\boldsymbol{y}_t \mid r_t^{(1)}, \boldsymbol{x}_t, \mathcal{D}_{1:t-1}) \, (1 - \kappa)}. \tag{37}$$

For details on the computation of (A.2: weighting), see Appendix A.3. For a detailed implementation of (M.2: auxvar) `RL`, (M.3: prior) `OUPR`, (A.2: weighting) `DA[1]`, and (A.1: posterior) `LG`, see Algorithm 4 in the Appendix.

For example, `RL[1]` is a runlength $r_t$ with a single hypothesis. We provide another example next.

CPL (changepoint location with retrospective membership): for $\psi_t = s_{1:t}$, the work of Nassar et al. (2022) evaluates the probability of past datapoints belonging in the current regime. In this scenario,

$$p(s_{1:t} \mid s_{1:t-1}, \mathcal{D}_{1:t-1}) = p(s_{1:t} \mid \mathcal{D}_{1:t-1}), \tag{38}$$

so that

$$p(s_{1:t} \mid \mathcal{D}_{1:t}) \propto p(s_{1:t} \mid \mathcal{D}_{1:t-1}) \, p(\boldsymbol{y}_t \mid \boldsymbol{x}_t, s_{1:t}, \mathcal{D}_{1:t-1}). \tag{39}$$

This method allows for exact computation by summing over all possible $2^t$ elements. However, to reduce the computational cost, they propose a discrete optimisation over possible values $\{\nu_t(s_{1:t}) : s_{1:t} \in \{0, 1\}^t\}$, where $\nu_t(s_{1:t}) = p(s_{1:t} \mid \mathcal{D}_{1:t})$. Then, the hypothesis with highest probability is stored and gets assigned a weight of one.

### 2.7.2 Continuous auxiliary variable (`CA`)

Here, we briefly discuss continuous auxiliary variables (`CA`). For some choices of $\psi_t$ and transition densities $p(\psi_t \mid \psi_{t-1}, \mathcal{D}_{1:t-1})$, computation of (28) becomes infeasible. In these scenarios, we use simpler approximations. We give an example below.

CPP (Changepoint probability with empirical Bayes estimate): for $\psi_t = \upsilon_t$, consider

$$p(\upsilon_t \mid \upsilon_{t-1}, \mathcal{D}_{1:t-1}) = p(\upsilon_t), \tag{40}$$

so that

$$p(v_t \mid \mathcal{D}_{1:t}) \propto p(v_t) \, p(\boldsymbol{y}_t \mid \boldsymbol{x}_t, v_t). \tag{41}$$

The work of Titsias et al. (2024) takes $\nu_t(v_t) = \delta(v_t - v_t^*)$, where $\delta$ is the Dirac delta function and $v_t^*$ is a point estimate centred at the maximum of the marginal posterior predictive likelihood:

$$v_t^* = \underset{v \in [0,1]}{\arg\max} \, p(\boldsymbol{y}_t \mid \boldsymbol{x}_t, v, \mathcal{D}_{1:t-1}). \tag{42}$$

In practice, (42) is approximated by taking gradient steps towards the minimum. This is a form of empirical Bayes approximation, since we compute the most likely value of the prior after marginalizing out $\boldsymbol{\theta}_t$. The work of Galashov et al. (2024) considers a modified configuration with choice of (M.2: auxvar) $\boldsymbol{v}_t \in (0,1)^m$.

## 3 Unified view of examples in the literature

Table 3 shows that many existing methods can be written as instances of BONE. Rather than specifying the choice of (M.1), we instead write the task for which it was designed, as discussed in Section 2.1. We will experimentally compare a subset of these methods in Section 4.

The methods presented in Table 3 can be directly applied to tackle any of the problems mentioned in Section 2.1. However, as choice of (M.1: model), we specify the task under which the configuration was introduced.[4]

| Reference | Task | M.2: auxvar | M.3: prior | A.1: posterior | A.2: weight |
|---|---|---|---|---|---|
| Kalman (1960) | filtering | C | LSSM | LG | DA[1] |
| Magill (1965) | filtering | ME | LSSM | LG | DA[K] |
| Chang & Athans (1978) | filtering | ME | LSSM | LG | CA |
| Chaer et al. (1997) | filtering | ME | LSSM | LG | DA[K] |
| Ghahramani & Hinton (2000) | SSSM | ME | Static | VB | CA |
| Adams & MacKay (2007) | seg. | RL | PR | Cj | DA[inf] |
| Fearnhead & Liu (2007) | seg. & preq. | CPT/ME | PR | Any | DA[inf] |
| Wilson et al. (2010) | seg. | RLCC | PR | Cj | DA[inf] |
| Fearnhead & Liu (2011) | seg. | CPT/ME | MMPR | Any | DA[inf] |
| Mellor & Shapiro (2013) | bandits | RL | PR | Cj | DA[inf] |
| Nguyen et al. (2017) | OCL | CPL | Sub | VB | DA[1] |
| Knoblauch & Damoulas (2018) | seg. | RL/ME | PR | Cj | DA[inf] |
| Kurle et al. (2019) | OCL | CPV | Sub | VB | DA[1] |
| Li et al. (2021) | OCL | CPL | MCI | VB | DA[inf] |
| Nassar et al. (2022) | bandits & OCL | CPV | Sub | LG | DA[1] |
| Liu (2023) | preq. | ME | C,LSSM | Any | DA[K] |
| Chang et al. (2023) | OCL | C | ACI | LG | DA[1] |
| Titsias et al. (2024) | OCL | CPP | OU | LG | CA |
| Galashov et al. (2024) | CL | CPP | OU | VB | CA |
| Abélès et al. (2024) | preq. | ME | LSSM | LG | DA[K] |
| RL[1]-OUPR* (ours) | any | RL | SPR | Any | DA[1] |

Table 3: List of methods ordered by publication date. The tasks are discussed in Section 2.1. We use the following abbreviations: SSSM means switching state space model; (O)CL means (online) continual learning; seg. means segmentation; preq. means prequential. Methods that consider two choices of (M.2: auxvar) are denoted by 'X/Y'. This corresponds to a double expectation in (6)—one for each choice of auxiliary variable.

## 4 Experiments

In this section we experimentally evaluate different algorithms within the BONE framework on a number of tasks.

---

[4]In general, the components of BONE can be thought as the building blocks for new methods. Some of these combinations would not be useful, but they can be employed nonetheless.

Each experiment consists of a *warmup* period where the hyperparameters are chosen, and a *deploy* period where sequential predictions and updates are performed. In each experiment, we fix the choice of measurement model $h$ (M.1: model) and posterior inference method (A.1: posterior), and then compare different methods with respect to their choice of (M.2: auxvar), (M.3: prior), and (A.2: weighting). For DA methods, we append the number of hypotheses in brackets to determine how many hypotheses are being considered. For example, RL[1]-PR denotes one hypothesis, RL[K]-PR denotes $K$ hypotheses, and RL[inf]-PR denotes all possible hypotheses. In all experiments, unless otherwise stated, we consider a single hypothesis for choices of DA. See Table 4 for the methods we compare.

| M.2-M.3 | Eq. | A.2 | Description | Sections |
|---|---|---|---|---|
| | | | static | |
| C-Static | (16) | - | This corresponds to the static case with a classical Bayesian update. This method does not assume changes in the environment. | 4.3.1, 4.3.2 |
| | | | abrupt changes | |
| RL-PR | (18) | DA[inf] | This approach, commonly referred to as Bayesian online change-point detection (**BOCD**), assumes that non-stationarity arises from independent blocks of time, each with stationary data. Estimates are made using data from the current block. See Appendix A.1 for more details. | 4.1.1, 4.1.2, 4.2, 4.3.1, 4.3.2, |
| WoLF+RL-PR* | (18) | DA[inf] | Special case of RL-PR with explicit choice of (M.1: model) which makes it robust to outliers. | 4.3.2 |
| | | | gradual changes | |
| CPP-OU | (14) | CA | Updates are done using a discounted mean and covariance according to the probability estimate that a change has occurred. | 4.1.1, 4.1.2, 4.2 |
| C-ACI | (15) | - | At each timestep, this method assumes that the parameters evolve according to a linear map $\mathbf{F}_t$, at a rate given by a known positive semidefinite covariance matrix $\mathbf{Q}_t$. | 4.1.1, 4.1.2, 4.2, |
| | | | abrupt & gradual changes | |
| RL-MMPR | (58) | DA[inf] | Modification of CPT-MMPR that assumes dependence between any two consecutive blocks of time and with choice of RL. This combination employs a moment-matching approach when evaluating the prior mean and covariance under a changepoint. See Appendix A.2 for more details. | 4.3.1 |
| RL-OUPR | (19) | DA[1] | Depending on the threshold parameter, updates involve either (i) a convex combination of the prior belief with the previous mean and covariance based on the estimated probability of a change (given the run length), or (ii) a hard reset of the mean and covariance, reverting them to prior beliefs. See Appendix A.3 for more details. | 4.1.1, 4.1.2, 4.2, 4.3.1 |

Table 4: List of methods we compare in our experiments. The first column, **M.2−M.3**, is defined by the choices of (M.2: auxvar) and (M.3: prior). The second column, **Eq.**, references the equation that define M.2–M.3. The third column, **A.2**, determines the choice of (A.2: weighting). The fourth column, **Description**, provides a brief summary of the method. The fifth column, **Sections**, shows the sections where the method is evaluated. The choice of (M.1: model) and (A.1: posterior) are defined on a per-experiment basis. (The only exception being WoLF+RL-PR). For (M.2: auxvar) the acronyms are as follows: RL means runlength, CPP means changepoint probability, C means constant, and CPT means changepoint timestep. For (M.3: prior) the acronyms are as follows: PR means prior reset, OU means Ornstein–Uhlenbeck, LSSM means linear state-space model, Static means full Bayesian update, MMR means moment-matched prior reset, and OUPR means Ornstein–Uhlenbeck and prior reset. We use the convention in Hušková (1999) for the terminology abrupt/gradual changes.

## 4.1 Prequential prediction

In this section, we give several examples of non-stationary prequential prediction problems.

### 4.1.1 Online regression for hour-ahead electricity forecasting

In this experiment, we consider the task of predicting the hour-ahead electricity load before and after the Covid pandemic. We use the dataset presented in Farrokhabadi et al. (2022), which has 31,912 observations;

each observation contains 7 features $\boldsymbol{x}_t$ and a single target variable $\boldsymbol{y}_t$. The 7 features correspond to pressure (kPa), cloud cover (%), humidity (%), temperature (C) , wind direction (deg), and wind speed (KmH). The target variable is the hour-ahead electricity load (kW). To preprocess the data, we normalise the target variable $\boldsymbol{y}_t$ by subtracting an exponentially weighted moving average (EWMA) mean with a half-life of 20 hours, then dividing the resulting series by an EWMA standard deviation with the same half-life. To normalise the features $\boldsymbol{x}_t$, we divide each by a 20-hour half-life EWMA. The features are lagged by one hour.

Our choice of measurement model $h$ is a two-hidden layer multilayered perceptron (MLP) with four units per layer and a ReLU activation function.

For this experiment, we consider `RL[1]-OUPR*` (our proposed method), `RL[1]-PR` (a classical method), `C-ACI` (a simple benchmark), and `CPP-OU` (a modern method). For computational convenience, we plug in a point-estimate (MAP estimate) of the neural network parameters when making predictions using $h$. More precisely, given $\psi_t$, we use $h(\boldsymbol{\theta}_t^*, \boldsymbol{x}_{t+1})$ to make a (conditional) prediction, where $\boldsymbol{\theta}_t^* = \arg\max_{\boldsymbol{\theta}} q(\boldsymbol{\theta}; \psi_t, \mathcal{D}_{1:t})$. For a fully Bayesian treatment of neural network predictions, see Immer et al. (2021); we leave the implementation of these approaches for future work.

The hyperparameters of each method are found using the first 300 observations (around 13 days) and deployed on the remainder of the dataset. Specifically, during the warmup period we tune the value of the probability of a changepoint for `RL[1]-OUPR*` and `RL[1]-PR`. For `C-ACI` we tune $\mathbf{Q}_t$, and for `CPP-OU` we tune the learning rate. See the open-source notebooks for more details.

In the top panel of Figure 3 we show the evolution of the target variable $\boldsymbol{y}_t$ between March 3 2020 and March 10 2020. The bottom panel of Figure 3 shows the 12-hour rolling mean absolute error (MAE) of predictions made by the methods. We see that there is a changepoint around March 7 2020 as pointed out in Farrokhabadi et al. (2022). This is likely due to the introduction of Covid lockdown rules. Among the methods considered, `C-ACI` and `RL[1]-OUPR*` adapt the quickest after the changepoint and maintain a low rolling MAE compared to `RL[1]-PR` and `CPP-OU`.

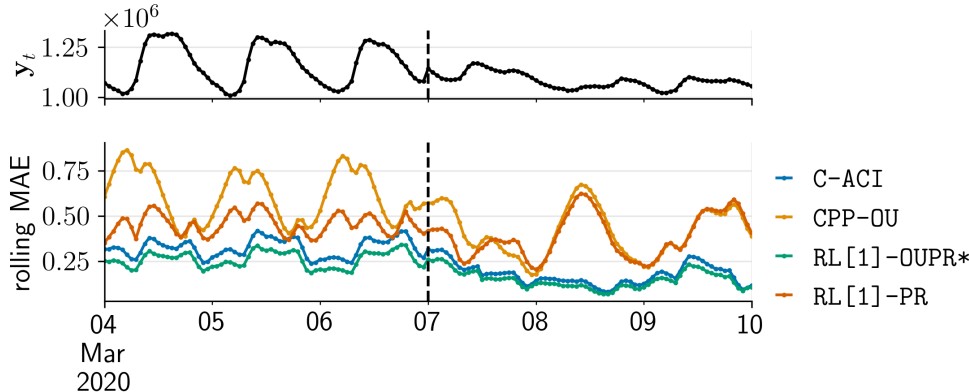

Figure 3: The **top panel** shows the target variable (electricity consumption) from March 1 2020 to March 12 2020. The **bottom panel** shows the twelve-hour rolling relative absolute error of predictions for the same time window. The dotted black line corresponds to March 7 2020, when Covid lockdown began.

Next, Figure 4, shows the forecasts made by each method between March 4 2020 and March March 8 2020. We observe a clear cyclical pattern before March 7 2020 but less so afterwards, indicating a change in daily electricity usage from diurnal to constant.

We also observe that `RL[1]-PR` and `CPP-OU` slow-down their rate of adaptation. One possible explanation of this behaviour is that the changes are not abrupt enough to be captured by the algorithms. To provide evidence for this hypothesis, Figure 5 shows, on the left $y$-axis, the predictions for `RL[1]-PR` and the target variable $\boldsymbol{y}_t$. On the right $y$-axis, we show the estimated runlength.

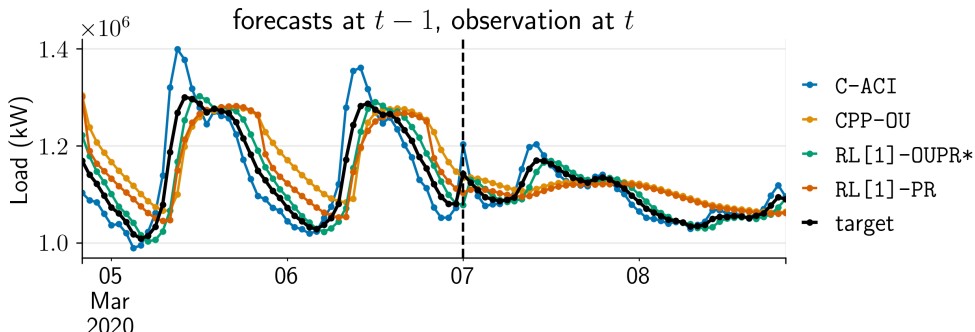

Figure 4: One day ahead electricity forecasting results for Figure 3. The dotted black line corresponds to March 7 2020.

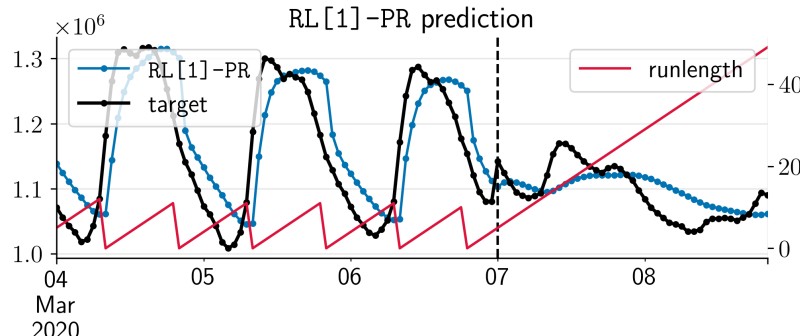

Figure 5: One day ahead electricity forecasting results for `RL[1]-PR` together with the target variable on the left y-axis, and the value for runlength (`RL`) on the right y-axis. We see that after the 7 March changepoint, the runlength monotonically increases, indicating a stationary regime.

We see that `RL[1]-PR` resets approximately twice every day until the time of the changepoint. After that, there is no evidence of a changepoint (as provided by the hyperparameters and the modelling choices), so `RL[1]-PR` does not reset which translates to less adaptation for the period to the right of the changepoint.

Finally, we compare the error of predictions made by the competing methods. This is quantified in Figure 6, which shows a box-plot of the five-day MAE for each of the competing methods over the whole dataset, from March 2017 to November 2020. Our new `RL[1]-OUPR*` method has the lowest MAE.

### 4.1.2 Online classification with periodic drift

In this section we study the performance of `C-ACI`, `CPP-OU`, `RL[1]-PR`, and `RL[1]-OUPR*` for the classification experiment of Section 6.2 in Kurle et al. (2019). More precisely, in this experiment $x_{t,i} \sim \text{Unif}[-3, 3]$ for $i \in \{1, 2\}$, $\boldsymbol{x}_t = (x_{t,1}, x_{t,2}) \in \mathbb{R}^2$, $y_t \sim \text{Bernoulli}(\sigma(\boldsymbol{\theta}_t^\mathsf{T} \boldsymbol{x}_t))$ with $\boldsymbol{\theta}_t^{(1)} = 10 \sin(5° t)$ and $\boldsymbol{\theta}_t^{(2)} = 10 \cos(5° t)$. Thus the unknown values of model parameters are slowly drifting deterministically according to sine and cosine functions. The timesteps go from 0 to 720.

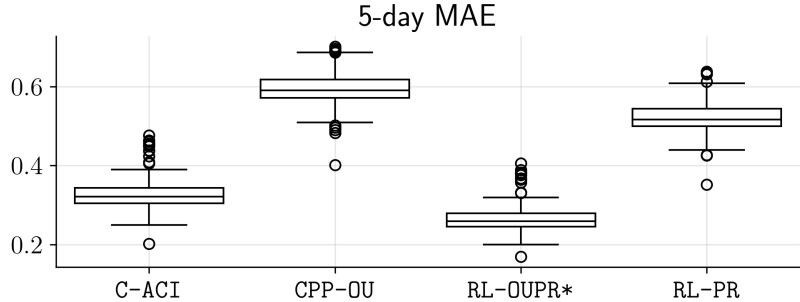

Figure 6: Distribution of the 5-day mean absolute error (MAE) for each of the competing methods on electricity forecasting over the entire period. For this calculation we split the dataset into consecutive buckets containing five days of data each, and for a given bucket we compute the average absolute error of the predictions and observations that fall within the bucket.

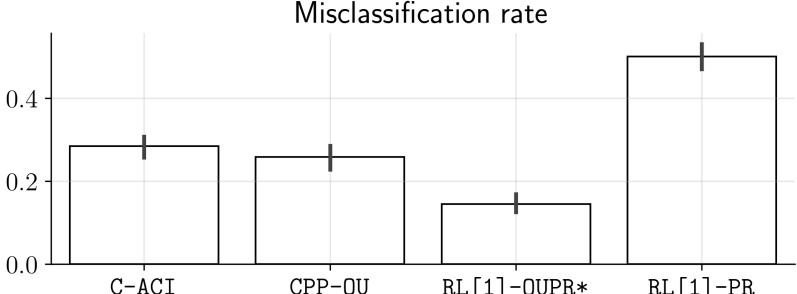

Figure 7: Misclassification rate of various methods on the online classification with periodic drift task.

Figure 7 summarises the results of the experiment where we show the misclassification rate (which is one minus the accuracy) for the competing methods. Our `RL[1]-OUPR*` method works the best, and signifcantly outperforms `RL[1]-PR`, since we use an OU drift process with a soft prior reset rather than assuming constant parameter with a hard prior rset.

We can improve the performance of `RL[K]-PR` if the number of hypotheses $K$ increases, and if we vary the changepoint probability threshold $\kappa$, as shown in Figure 8. However, even then the performance of this method still does not match our method.

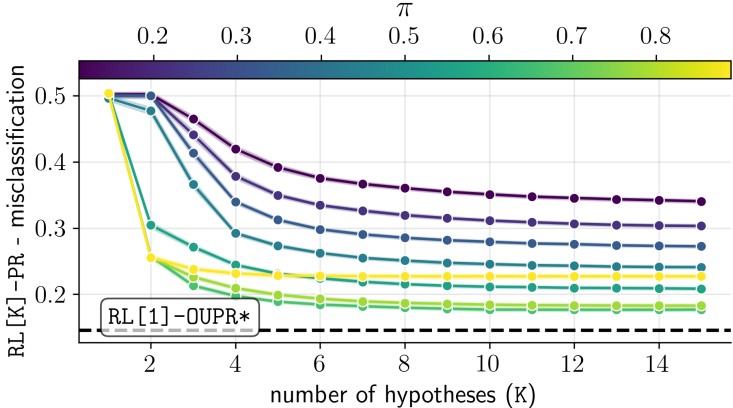

Figure 8: Accuracy of predictions for `RL[1]-PR` as a function of the number of hypothesis and the prior probability of a changepoint $\kappa$. The black dotted line is the performance of `RL[1]-OUPR*` reported in Figure 7.

### 4.1.3 Online classification with drift and jumps

In this section we study the performance of `C-ACI`, `CPP-OU`, `RL[1]-PR`, and `RL[1]-OUPR*` for an experiment with drift and sudden changes. More precisely, we assume that the parameters of a logistic regression problem evolve according to

$$
\boldsymbol{\theta}_t = \begin{cases} \boldsymbol{\theta}_{t-1} + \boldsymbol{\epsilon}_t & \text{w.p. } 1 - p_\epsilon, \\ \mathcal{U}[-2,2]^2 & \text{w.p. } p_\epsilon, \end{cases} \tag{43}
$$

with $p_\epsilon = 0.01$, $\boldsymbol{\theta}_0 \sim \mathcal{U}[-2,2]^2$, and $\boldsymbol{\epsilon}_t$ is a zero-mean distributed random vector with isotropic covariance matrix $(0.01)^2 \mathbf{I}_2$ (where $\mathbf{I}_2$ is a $2 \times 2$ identity matrix). Intuitively, this experiment has model parameters that drift slowly with occasional abrupt changes (at a rate of 0.01).

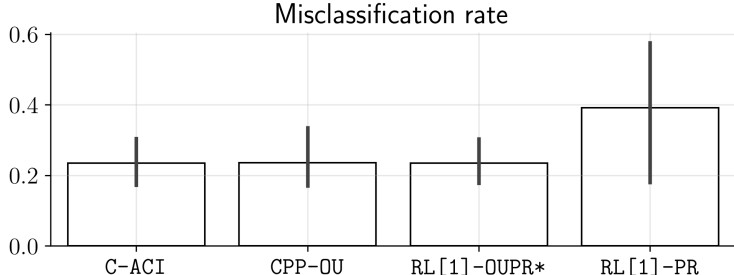

Figure 9: Misclassification rate of various methods on the online classification with drift and jumps task.

Figure 9 shows the missclasification rate among the competing methods. We observe that `C-ACI`, `CPP-OU`, and `RL[1]-OUPR*` have comparable performance, whereas `RL[1]-PR` is the method with highest misclassification rate.

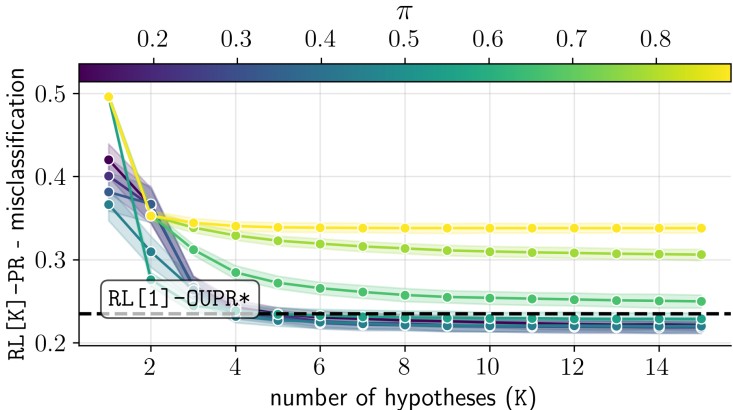

Figure 10: Accuracy of predictions for `RL[K]-PR` as a function of the number of hypotheses (`K`) and the probability of a changepoint $\kappa$. The black dotted line is the performance of `RL[1]-OUPR*` reported in Figure 9.

To explain this behaviour, Figure 10 shows the performance of `RL[K]-PR` as a function of number of hypotheses and prior probability of a changepoint $\kappa$. We observe that up to three hypotheses, the lowest misclassification error of `RL[K]-PR` is higher than that of `RL[1]-OUPR*`, which only considers one hypothesis. However, as we increase the number of hypotheses, the best performance for `RL[K]-PR` obtains a lower misclassification rate than `RL[1]-OUPR*`. This is in contrast to the results in Figure 5. Here, we see that with more hypotheses `RL[K]-PR` outperforms our new method at the expense of being more memory intensive.

## 4.2 Contextual bandits

In this section, we study the performance of `C-ACI`, `CPP-OU`, `RL[1]-PR`, and `RL[1]-OUPR*` for the simple Bernoulli bandit from Section 7.3 of Mellor & Shapiro (2013). More precisely, we consider a multi-armed bandit problem with 10 arms, 10,000 steps per simulation, and 100 simulations. The payoff of a given arm is the outcome of a Bernoulli random variable with unknown probability $\theta_t = \min\{\max\{\theta_{t-1} + 0.03\,Z_t, 0\}, 1\}$ for $\{Z_t\}_{t \in \{1,2,\dots,10,000\}}$ independent and identically distributed standard normal random variables. We take $\theta_0 \sim \text{Unif}[0,1]$ and use the same formulation for all ten arms with independence across arms. The observations are the rewards and there are no features (non-contextual).

The idea of using `RL[1]-PR` in multi-armed bandits problems was introduced in Mellor & Shapiro (2013). With this experiment, we extend the concept to other members of the BONE framework. We use Thompson sampling for each of the competing methods. Figure 11 shows the regret of using `C-ACI`, `CPP-OU`, `RL[1]-PR`, and `RL[1]-OUPR*` for the above multi-armed bandits problem. The results we obtain are similar to those of Section 4.1.2. This is because both problems have a similar drift structure.

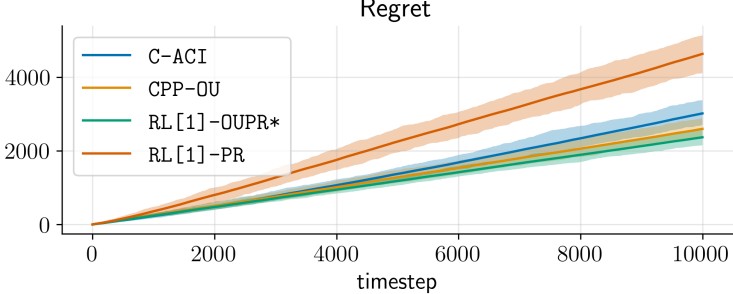

Figure 11: Regret of competing methods on the contextual bandits task. Confidence bands are computed with one hundred simulations.

### 4.3 Segmentation and prediction

In this section, we evaluate methods both in terms of their ability to "correctly" segment the observed output signal, and to do one-step-ahead predictions. Note that by "correct segmentation", we mean one that matches the ground truth data generating process. This metric can only be applied to synthetic data.

#### 4.3.1 Autoregression with dependence across the segments

In this experiment, we consider the synthetic autoregressive dataset introduced in Section 2 of Fearnhead & Liu (2011), consisting of a set of one dimensional polynomial curves that are constrained to match up at segmentation boundaries, as shown in the top left of Figure 12.

We compare the performance of the three methods in the previous subsection. For this experiment, we employ a probability of a changepoint $\kappa = 0.01$. Since this dataset has dependence of the parameters across segments, we allow for the choice of (M.1: model) to be influenced by the choice of (M.2: auxvar), i.e., our choice of model is given by $h(\boldsymbol{\theta}_t; \psi_t, \boldsymbol{x}_t)$. For this experiment, we take (M.2: auxvar) to be RL and our choice of (M.1: model) becomes

$$h(\boldsymbol{\theta}_t; r_t, \boldsymbol{x}_{1:t}) = \boldsymbol{\theta}_t^\mathsf{T}\, \boldsymbol{h}(\boldsymbol{x}_{1:t}, r_t), \tag{44}$$

with $\boldsymbol{h}(\boldsymbol{x}_{1:t}, r_t) = [1, \Delta, \Delta^2]$, $\Delta = (x_t - x_{r_t})$, and $x_{r_t} \geq x_t$. Intuitively this represents a quadratic curve fit to the beginning $x_{r_t}$ and end points $x_t$ of the current segment. Given the form of (M.1: model) in (44), here we do not consider C-ACI nor CPP-OU. Instead, we use runlength with moment-matching prior reset, i.e., RL-MMPR (see Table 4) which was designed for segmentation with dependence.

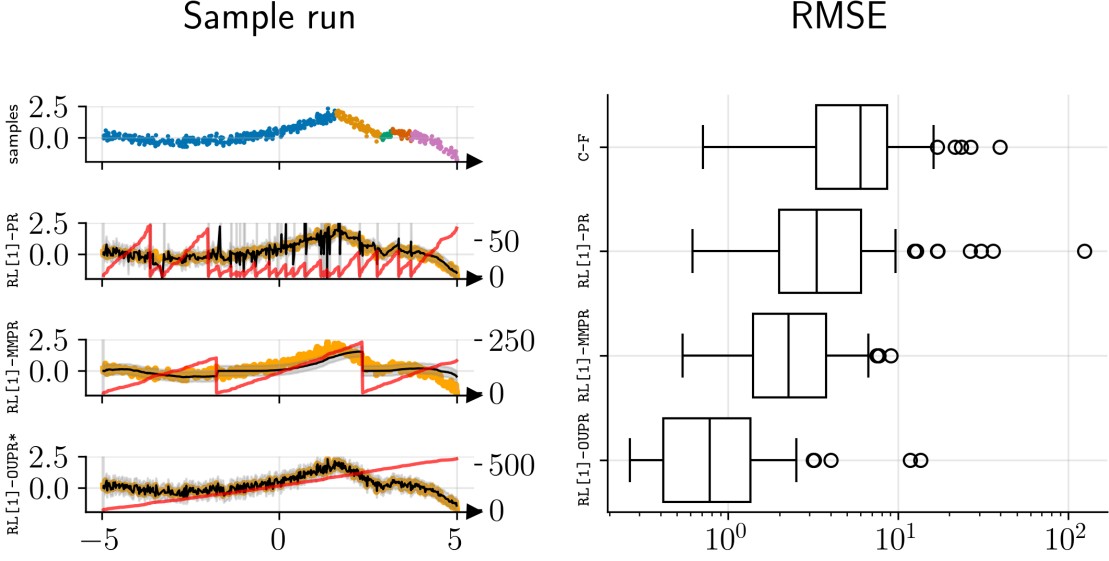

Figure 12: The **left panel** shows a sample run of the piecewise polynomial regression with dependence across segments. The $x$-axis is for the features, the (left) $y$-axis is for measurements together with the estimations made by RL[1]-PR, RL-MMPR, and RL[1]-OUPR*, the (right) $y$-axis is for the value of $r_t$ under each model. The orange line denotes the true data-generating process and the red line denotes the value of the hypothesis RL. The **right panel** shows the RMSE of predictions over 100 trials.

Figure 12 shows the results. On the right, we observe that RL[1]-OUPR* has the lowest RMSE. On the left, we plot the predictions of each method, so we can visualise the nature of their errors. For RL[1]-PR, the spikes occur because the method has many false positive beliefs in a changepoint occurring, and this causes breaks in the predictions due the explicit dependence of $h$ on $r_t$ and the hard parameter reset upon changepoints. For RL-MMPR, the slow adaptation (especially when $x_t \in [1, 5]$) is because the method does not adjust beliefs as quickly as it should. Our RL[1]-OUPR* method strikes a good compromise.

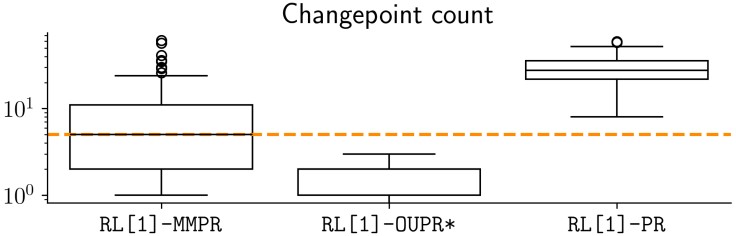

Figure 13: Count of changepoints over an experiment for 100 trials. The orange line shows the true number of changepoints for all trials.

Figure 13 shows the distribution (over 100 simulations) of the number of detected changepoints, i.e., instances where $\nu_t(r_t)$ with $r_t = 0$ is the highest. We observe that superior predictive performance in Figure 12 does not necessarily translate to a better segmentation capability. For example, the distribution produced by `RL-MMPR` sits around the actual number of changepoints (better at segmenting) whereas `RL[1]-OUPR*`, which is detecting far fewer changepoints, is the best performing prediction method. This reflects the discrepancy between the objectives of segmentation and prediction. For a more thorough analysis and evaluation of changepoint detection methods on time-series data, see Van den Burg & Williams (2020).

### 4.3.2 Non-stationary heavy-tailed regression with `DA[inf]`

It is well-known that the combination `RL-PR` is sensitive to outliers if the choice of (M.1: model) is misspecified, since an observation that is "unusual" may trigger a changepoint unnecessarily. As a consequence, various works have proposed outlier-robust variants to the `RL[inf]-PR` for segmentation (Knoblauch et al., 2018; Fearnhead & Rigaill, 2019; Altamirano et al., 2023; Sellier & Dellaportas, 2023) and for filtering (Reimann, 2024). In what follows, we show how we can easily accomodate robust methods into the BONE framework by changing the way we compute the likelihood and/or posterior. In particular, we consider the WoLF-IMQ method of Duran-Martin et al. (2024). We use WoLF-IMQ because it is a provably robust algorithm and it is a straightforward modification of the linear Gaussian posterior update equations. We denote `RL[inf]-PR` with (A.1: posterior) taken to be `LG` as `LG+RL[inf]-PR` and `RL[inf]-PR` with (A.1: posterior) taken to be WoLF-IMQ as `WoLF+RL[inf]-PR*`.

To demonstrate the utility of a robust method, we consider a piecewise linear regression model with Student-$t$ errors, where the measurement are sampled according to $\boldsymbol{x}_t \sim \mathcal{U}[-2,2]$, $\boldsymbol{y}_t \sim \mathrm{St}\big(\phi(\boldsymbol{x}_t)^\intercal\boldsymbol{\theta}_t, 1, \ 2.01\big)$ a Student-$t$ distribution with location $\phi(\boldsymbol{x}_t)^\intercal\boldsymbol{\theta}_t$, scale 1, degrees of freedom 2.01, and $\phi(\boldsymbol{x}_t) = (1, x, x^2)$. At every timestep, the parameters take the value

$$\boldsymbol{\theta}_t = \begin{cases} \boldsymbol{\theta}_{t-1} & \text{w.p. } 1 - p_\epsilon, \\ \mathcal{U}[-3,3]^3 & \text{w.p. } p_\epsilon, \end{cases} \tag{45}$$

with $p_\epsilon = 0.01$, and $\boldsymbol{\theta}_0 \sim \mathcal{U}[-3,3]^3$. Intuitively, at each timestep, there is probability $p_\epsilon$ of a changepoint, and conditional on a changepoint occurring, the each of the entries of the new parameters $\boldsymbol{\theta}_t$ are sampled from a uniform in $[-3,3]$. Figure 14 shows a sample data generated by this process.

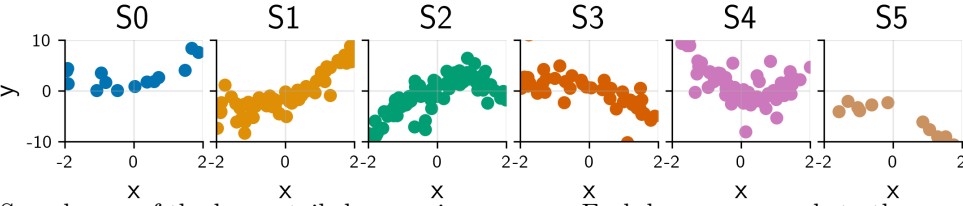

Figure 14: Sample run of the heavy-tailed-regression process. Each box corresponds to the samples within a segment.

To process this data, our choice of (M.1: model) is $h(\boldsymbol{\theta}_t, \boldsymbol{x}_t) = \boldsymbol{\theta}_t^\intercal \, \phi(\boldsymbol{x}_t)$ with

$$\ell(\boldsymbol{y}_t; \boldsymbol{\theta}_t, \boldsymbol{x}_t) = -W^2\big(\boldsymbol{y}_t, h(\boldsymbol{\theta}_t, \boldsymbol{x}_t)\big) \log \mathcal{N}(\boldsymbol{y}_t \,|\, h(\boldsymbol{\theta}_t, \boldsymbol{x}_t), 1.0), \tag{46}$$

a weighted Gaussian log-likelihood and $W(u, z) = (1 + \frac{(u-z)^2}{c^2})^{-1/2}$ the inverse multi-quadratic (IMQ) function with soft threshold value $c = 4$, representing four standard deviations of tolerance to outliers. Here $u, z \in \mathbb{R}$.

The left panel in Figure 15 shows the rolling mean (with a window of size 10) of the RMSE for `LG+RL[inf]-PR`, `WoLF+RL[inf]-PR*`, and `LG+C-Static`. The right panel in Figure 15 shows the distribution of the RMSE for all methods after 30 trials.

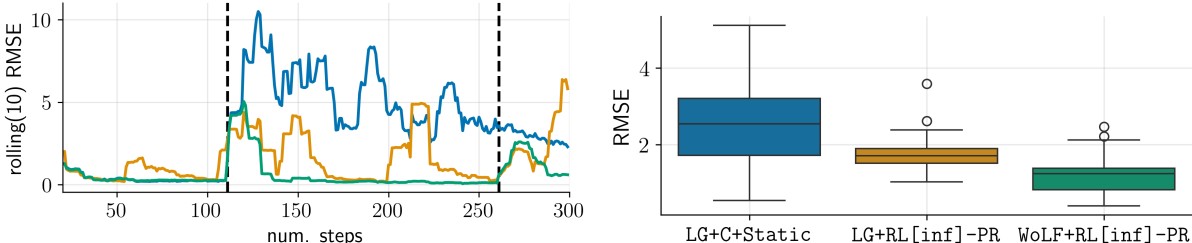

Figure 15: The **left panel** shows the rolling RMSE using a window of the 10 previous observations. The **right panel** shows the distribution of final RMSE over 30 runs. The vertical dotted line denotes a change in the true model parameters.

The left panel of Figure 15 shows that `LG+C-Static` has a lower rolling RMSE error than `LG+RL[inf]-PR` up to first changepoint (around 100 steps). The performance of `LG+C-Static` significantly deteriorates afterwards. Next, `LG+RL[inf]-PR` wrongly detects changepoints and resets its parameters frequently. This results in periods of increased rolling RMSE. Finally, `WoLF+RL[inf]-PR*` has the lowest error among the methods. After the regime change, its error increases at a similar rate to the other methods, however, it correctly adapts to the regime and its error decreases soon after the changepoint.

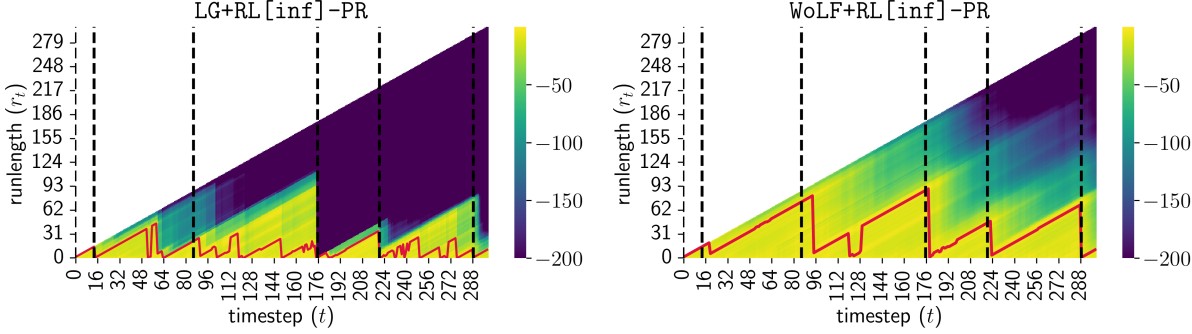

Figure 16: Segmentation of the non-stationary linear regression problem. The left panel shows the segmentation done by `LG+RL[inf]-PR`. The right panel shows the segmentation done by `WoLF+RL[inf]-PR*`. The $x$-axis is the timestep $t$, the $y$-axis is the runlength $r_t$ (note that it is always the case that $r_t \leq t$), and the color bar shows the value $\log p(r_t \,|\, \boldsymbol{y}_{1:t})$. The red line in either plot is the trajectory of the mode, i.e., the set $r_{1:t}^* = \{\arg\max_{r_1} p(r_1 \,|\, \mathcal{D}_1), \ldots, \arg\max_{r_t} p(r_t \,|\, \mathcal{D}_{1:t})\}$. Note that the non-robust method (left) oversegments the signal. See this url for a video comparison between `LG+RL[inf]-PR` and `WoLF+RL[inf]-PR*`.

Figure 16 shows the posterior belief of the value of the runlength using `LG+RL[inf]-PR` and `WoLF+RL[inf]-PR*`. The constant reaction to outliers in the case of `LG+RL[inf]-PR` means that the parameters keep resetting back to the initial prior belief. As a consequence, the RMSE of `LG+RL[inf]-PR` deteriorates. On the other hand, `WoLF+RL[inf]-PR*` resets less often, and accurately adjusts to the regime changes when they do happen. This results in the lowest RMSE among the three methods.

## 5 Conclusions

We introduced a unified Bayesian framework to perform online predictions in non-stationary environments, and showed how it covers many prior works. We also used our framework to design a new method, `RL[1]-OUPR*`, which is suited to tackle prediction problems when the observations exhibit both abrupt and gradual changes. In future work, we aim to investigate other novel variants and applications to newer model architectures such as transformers and graph neural networks (Moreno-Pino et al., 2024; Arroyo et al., 2025).

## Aknowledgments

We thank Matias Altamirano, François-Xavier Briol, Patrick Chang, Alex Galashov, Matt Jones, Jeremias Knoblauch, and Radford Neal for insightful comments and discussions on earlier versions of this paper.

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

# A Worked examples for BONE methods

In this section, we provide a detailed calculation of $\nu_t(\psi_t)$ for some choices of $\psi_t$. We consider a choice of (M.1: model) to be linear Gaussian with known observation variance $\mathbf{R}_t$, i.e.,

$$p(\boldsymbol{y}_t \,|\, \boldsymbol{\theta}, \boldsymbol{x}_t) = \mathcal{N}(\boldsymbol{y}_t \,|\, \boldsymbol{x}_t^\mathsf{T} \boldsymbol{\theta}_t, \mathbf{R}_t). \tag{47}$$

## A.1 Runlength with prior reset (RL-PR)

### A.1.1 Unbounded number of hypotheses RL[inf]-PR

The work in Adams & MacKay (2007) takes $\psi_t = r_t$ to be the runlength, with $r_t \in \{0, 1, \dots, t\}$, that that counts the number of steps since the last changepoint. Assume the runlength follows the dynamics (30). We consider $\nu_t(r_t) = p(r_t|\mathcal{D}_{1:t})$ such that

$$p(r_t \,|\, \mathcal{D}_{1:t}) = \frac{p(r_t, \mathcal{D}_{1:t})}{\sum_{\hat{r}_t=0}^{t} p(\hat{r}_t, \mathcal{D}_{1:t})}, \tag{48}$$

for $r_t \in \{0, \dots, t\}$. The RL-PR method estimates $p(r_t, \mathcal{D}_{1:t})$ for all $r_t \in \{0, \dots, t\}$ at every timestep. To estimate this value recursively, we sum over all possible previous runlengths as follows

$$
\begin{aligned}
&p(r_t, \mathcal{D}_{1:t}) \\
&= \sum_{r_{t-1}=0}^{t-1} p(r_t, r_{t-1}, \mathcal{D}_{1:t-1}, \mathcal{D}_t) \\
&= \sum_{r_{t-1}=0}^{t-1} p(r_{t-1}, \mathcal{D}_{1:t-1}) \, p(r_t \,|\, r_{t-1}, \mathcal{D}_{1:t-1}) \, p(\boldsymbol{y}_t \,|\, r_t, r_{t-1}, \boldsymbol{x}_t, \mathcal{D}_{1:t-1}) \\
&= p(\boldsymbol{y}_t \,|\, r_t, \boldsymbol{x}_t, \mathcal{D}_{1:t-1}) \sum_{r_{t-1}=0}^{t-1} p(r_{t-1}, \mathcal{D}_{1:t-1}) p(r_t \,|\, r_{t-1}).
\end{aligned} \tag{49}
$$

In the last equality, there are two implicit assumptions, (i) the runlength at time $t$ is conditionally independent of the data $\mathcal{D}_{1:t-1}$ given the runlength at time $t-1$, and (ii) the model is Markovian in the runlength, that is, conditioned on $r_t$, the value of $r_{t-1}$ bears no information. Mathematically, this means that (i) $p(r_t \,|\, r_{t-1}, \mathcal{D}_{1:t-1}) = p(r_t \,|\, r_{t-1})$ and (ii) $p(\boldsymbol{y}_t \,|\, r_t, r_{t-1}, \mathcal{D}_{1:t-1}) = p(\boldsymbol{y}_t \,|\, r_t, \mathcal{D}_{1:t-1})$. From (49), we observe there are only two possible scenarios for the value of $r_t$. Either $r_t = 0$ or $r_t = r_{t-1} + 1$ with $r_{t-1} \in \{0, \dots, t-1\}$. Thus, $p(r_t, \mathcal{D}_{1:t})$ becomes

$$
\begin{aligned}
p(r_t, \mathcal{D}_{1:t}) &= p(\boldsymbol{y}_t \,|\, r_t, \boldsymbol{x}_t, \mathcal{D}_{1:t-1}) \, p(r_{t-1}, \mathcal{D}_{1:t-1}) \, p(r_t \,|\, r_{t-1}) && \text{if } r_t \geq 1 \\
p(r_t, \mathcal{D}_{1:t}) &= p(\boldsymbol{y}_t \,|\, r_t, \boldsymbol{x}_t, \mathcal{D}_{1:t-1}) \sum_{r_{t-1}=0}^{t-1} p(r_{t-1}, \mathcal{D}_{1:t-1}) \, p(r_t \,|\, r_{t-1}) && \text{if } r_t = 0 \,.
\end{aligned} \tag{50}
$$

The joint density (50) considers two possible scenarios: either we stay in a regime considering the past $r_t \geq 1$ observations, or we are in a new regime, in which $r_t = 0$. Finally, note that (50) depends on three terms: (i) the transition probability $p(r_t \,|\, r_{t-1})$, which it is assumed to be known, (ii) the previous log-joint

$p(r_{t-1}, \mathcal{D}_{1:t-1})$, with $r_{t-1} \in \{0, 1, \ldots, t-1\}$, which is estimated at the previous timestep, and (iii) the prior predictive density

$$p(\boldsymbol{y}_t \,|\, r_t, \boldsymbol{x}_t, \mathcal{D}_{1:t-1}) = \int p(\boldsymbol{y}_t \,|\, \boldsymbol{\theta}_t, \boldsymbol{x}_t)\, p(\boldsymbol{\theta}_t \,|\, r_t, \mathcal{D}_{1:t-1}) \mathrm{d}\boldsymbol{\theta}_t. \tag{51}$$

For a choice of (M.1: model) given by (47) and a choice of (M.3: prior) given by (18), the posterior predictive (51) takes the form.

$$
\begin{aligned}
p(\boldsymbol{y}_t \,|\, r_t, \boldsymbol{x}_t, \mathcal{D}_{1:t-1}) &= \int \mathcal{N}\left(\boldsymbol{y}_t \,|\, \boldsymbol{x}_t^\mathsf{T} \boldsymbol{\theta}_t, \mathbf{R}_t\right) \mathcal{N}\left(\boldsymbol{\theta}_t \,|\, \boldsymbol{\mu}_{t-1}^{(r_t)}, \boldsymbol{\Sigma}_{t-1}^{(r_t)}\right) \mathrm{d}\boldsymbol{\theta}_t \\
&= \mathcal{N}\left(\boldsymbol{y}_t \,|\, \boldsymbol{x}_t^\mathsf{T} \boldsymbol{\mu}_{t-1}^{(r_t)},\, \boldsymbol{x}_t^\mathsf{T} \boldsymbol{\Sigma}_{t-1}^{(r_t)} \boldsymbol{x}_t + \mathbf{R}_t\right),
\end{aligned}
\tag{52}
$$

with $r_t \in \{0, \ldots, t-1\}$. Here, $\left(\boldsymbol{\mu}_{t-1}^{(r_t)}, \boldsymbol{\Sigma}_{t-1}^{(r_t)}\right)$ are the posterior mean and covariance at time $t-1$ built using the last $r_t \geq 1$ observations. If $r_t = 0$, then $(\boldsymbol{\mu}_{t-1}^{(r_t)}, \boldsymbol{\Sigma}_{t-1}^{(r_t)}) = (\boldsymbol{\mu}_0, \boldsymbol{\Sigma}_0)$.

### A.1.2 Bounded number of hypotheses `RL[K]-PR`

If we maintain a set of $K$ possible hypotheses, then $\boldsymbol{\Psi}_t = \{r_{t-1}^{(1)}, \ldots, r_{t-1}^{(K)}\} \in \{0, \ldots, t-1\}^K$ is a collection of $K$ unique runlengths obtained at time $t-1$. Next, (49) takes the form

$$p(r_t, \mathcal{D}_{1:t}) = p(\boldsymbol{y}_t \,|\, r_t, \boldsymbol{x}_t, \mathcal{D}_{1:t-1})\, p(r_{t-1}, \mathcal{D}_{1:t-1})\, p(r_t \,|\, r_{t-1}) \qquad \text{if } r_t \geq 1, \tag{53}$$

$$p(r_t, \mathcal{D}_{1:t}) = p(\boldsymbol{y}_t \,|\, r_t, \boldsymbol{x}_t, \mathcal{D}_{1:t-1}) \sum_{r_{t-1} \in \boldsymbol{\Psi}_{t-1}} p(r_{t-1}, \mathcal{D}_{1:t-1})\, p(r_t \,|\, r_{t-1}) \qquad \text{if } r_t = 0. \tag{54}$$

Here, we have that either $r_t = r_{t-1} + 1$ when $r_{t-1} \in \boldsymbol{\Psi}_{t-1}$ or $r_t = 0$. After computing $p(r_t, \mathcal{D}_{1:t})$ for all $K+1$ possibles values of $r_t$, a choice is made to keep $K$ hypotheses. For timesteps $t \leq K$, we evaluate all possible hypotheses until $t > K$.

Algorithm 2 shows an update step under this process when we maintain a set of $K$ possible hypotheses.

### A.2 Runlength with moment-matched prior reset (`RL-MMPR`)

Here, we consider a modified version of the method introduced in Fearnhead & Liu (2011). We consider the choice of `RL` and adjust the choice of (M.3: prior) for `RL-PR` introduced in Appendix A.1 whenever $r_t = 0$. In this combination, for $r_t = 0$, we take $\tau(\boldsymbol{\theta}_t \,|\, r_t, \mathcal{D}_{1:t-1}) = p(\boldsymbol{\theta}_t \,|\, r_t, \mathcal{D}_{1:t-1})$. Next

$$
\begin{aligned}
p(\boldsymbol{\theta}_t \,|\, r_t, \mathcal{D}_{1:t-1}) &= \sum_{r_{t-1}=0}^{t-1} p(\boldsymbol{\theta}_t, r_{t-1} \,|\, r_t, \mathcal{D}_{1:t-1}) \\
&= \sum_{r_{t-1}=0}^{t-1} p(r_{t-1} \,|\, \mathcal{D}_{1:t-1})\, p(r_t \,|\, r_{t-1})\, p(\boldsymbol{\theta}_t \,|\, r_t, r_{t-1}, \boldsymbol{y}_{1:t-1}) \\
&= \sum_{r_{t-1}=0}^{t-1} p(r_{t-1} \,|\, \mathcal{D}_{1:t-1})\, p(r_t \,|\, r_{t-1})\, \mathcal{N}(\boldsymbol{\theta}_t \,|\, \boldsymbol{\mu}_{t-1}^{(r_{t-1})}, \boldsymbol{\Sigma}_{t-1}^{(r_{t-1})}).
\end{aligned}
\tag{55}
$$

Because (55) is a mixture model, we choose a conditional prior to be Gaussian that approximates the first two moments. We obtain

$$\mathbb{E}[\boldsymbol{\theta}_t \,|\, r_t, \boldsymbol{y}_{1:t-1}] = \sum_{r_{t-1}=0}^{t-1} p(r_{t-1} \,|\, \mathcal{D}_{1:t-1})\, p(r_t \,|\, r_{t-1})\, \boldsymbol{\mu}_{t-1}^{(r_{t-1})} \tag{56}$$

for the first moment, and

$$\mathbb{E}[\boldsymbol{\theta}_t\, \boldsymbol{\theta}_t^\mathsf{T} \,|\, r_t, \boldsymbol{y}_{1:t-1}] \sum_{r_{t-1}=0}^{t-1} p(r_{t-1} \,|\, \mathcal{D}_{1:t-1})\, p(r_t \,|\, r_{t-1}) \left(\boldsymbol{\Sigma}_{t-1}^{(r_{t-1})} + \boldsymbol{\mu}_{t-1}^{(r_{t-1})} \boldsymbol{\mu}_{t-1}^{(r_{t-1})\mathsf{T}}\right) \tag{57}$$

for the second moment. The conditional prior mean and prior covariance under $r_t = 0$ take the form

$$
\begin{aligned}
\boldsymbol{\mu}_t^{(0)} &= \mathbb{E}[\boldsymbol{\theta}_t \,|\, r_t, \boldsymbol{y}_{1:t-1}], \\
\boldsymbol{\Sigma}_t^{(0)} &= \mathbb{E}[\boldsymbol{\theta}_t \,\boldsymbol{\theta}_t^\intercal \,|\, r_t, \boldsymbol{y}_{1:t-1}] - (\mathbb{E}[\boldsymbol{\theta}_t \,|\, r_t, \boldsymbol{y}_{1:t-1}]) \, (\mathbb{E}[\boldsymbol{\theta}_t \,|\, r_t, \boldsymbol{y}_{1:t-1}])^\intercal .
\end{aligned}
\tag{58}
$$

Algorithm 3 shows an update step under this process when we maintain a set of $K$ possible hypotheses.

### A.3 Runlength with OU dynamics and prior reset (`RL[1]-OUPR*`)

In this section, we provide pseudocode for the new hybrid method we propose. Specifically, our choices in BONE are: `RL[1]-OUPR*` for (M.2: auxvar) and (M.3: prior), `LG` for (A.1: posterior), and `DA[1]` for (A.2: weighting). Because of our choice of (A.2: weighting), `RL[1]-OUPR*` considers a single hypothesis (or runlength) which, at every timestep, is either increased by one or set back to zero, according to the probability of a changepoint and a threshold $\epsilon \in (0,1)$.

In essence, `RL[1]-OUPR*` follows the logic behind `RL[1]-PR` introduced in Section A.1 with $K = 1$ hypothesis and different choice of (M.3: prior). To derive the algorithm for `RL[1]-OUPR*` at time $t > 1$, suppose $r_{t-1}$ is available (the only hypothesis we track). Denote by $r_t^{(1)}$ the hypothesis of a runlength increase, i.e., $r_t = r_{t-1} + 1$ and denote by $r_t^{(0)}$ the hypothesis of a runlenght reset, i.e., $r_t = 0$. The probability of a runlength increase under a single hypothesis takes the form:

$$
\begin{aligned}
\nu_t(r_t^{(1)}) &= p(r_t^{(1)} \,|\, \mathcal{D}_{1:t}) \\
&= \frac{p(r_t^{(1)}, \mathcal{D}_{1:t})}{p(r_t^{(1)}, \mathcal{D}_{1:t}) + p(r_t^{(0)}, \mathcal{D}_{1:t})} \\
&= \frac{p(\boldsymbol{y}_t \,|\, r_t^{(1)}, \boldsymbol{x}_t, \mathcal{D}_{1:t-1}) \, p(r_{t-1}, \mathcal{D}_{1:t-1}) \, (1 - \kappa)}{p(\boldsymbol{y}_t \,|\, r_t^{(0)}, \boldsymbol{x}_t, \mathcal{D}_{1:t-1}) \, p(r_{t-1}, \mathcal{D}_{1:t-1}) \, \kappa + p(\boldsymbol{y}_t \,|\, r_t^{(1)}, \boldsymbol{x}_t, \mathcal{D}_{1:t-1}) \, p(r_{t-1}, \mathcal{D}_{1:t-1}) \, (1 - \kappa)} \\
&= \frac{p(\boldsymbol{y}_t \,|\, r_t^{(1)}, \boldsymbol{x}_t, \mathcal{D}_{1:t-1}) \, (1 - \kappa)}{p(\boldsymbol{y}_t \,|\, r_t^{(0)}, \boldsymbol{x}_t, \mathcal{D}_{1:t-1}) \, \kappa + p(\boldsymbol{y}_t \,|\, r_t^{(1)}, \boldsymbol{x}_t, \mathcal{D}_{1:t-1}) \, (1 - \kappa)} .
\end{aligned}
\tag{59}
$$

where $\kappa = p(r_t \,|\, r_{t-1})$ with $r_t = 0$ is the prior probability of a changepoint and and $1 - \kappa = p(r_t \,|\, r_{t-1}$ with $r_t = r_{t-1} + 1$ is the probability of continuation of the current segment.

Next, we use $\nu_t(r_t)$ to decide whether to update our parameters or reset them according to a prior belief according to some threshold $\epsilon$. This implements our choice of (M.3: prior) given in (19) and (20). Because we maintain a single hypothesis, the weight at the end of the update step is set to 1. Algorithm 4 shows an update step for `RL[1]-OUPR*` under the choice of (M.1: model) given by (47).

### A.4 Changepoint location with multiplicative covariance inflation `CPL-MCI`

The work in Li et al. (2021) takes $\psi_t = s_{1:t}$ to be a $t$-dimensional vector where the $i$-th element is a binary vector that determines a changepoint at time $t$. Then, the sum of the entries of $s_{1:t}$ represents the total number of changepoints up to, and including, time $t$.

We take $\nu_t(s_{1:t}) = p(s_{1:t}|\mathcal{D}_{1:t})$, which is recursively expressed as

$$
\begin{aligned}
p(s_{1:t} \,|\, \mathcal{D}_{1:t}) &= p(s_t, s_{1:t-1} \,|\, \boldsymbol{y}_t, \boldsymbol{x}_t, \mathcal{D}_{1:t-1}) \\
&= p(s_{1:t-1} \,|\, \mathcal{D}_{1:t-1}) p(s_t \,|\, s_{1:t-1}, \boldsymbol{x}_t, \boldsymbol{y}_t, \mathcal{D}_{1:t-1}).
\end{aligned}
\tag{60}
$$

Here, $p(s_{1:t-1} \,|\, \mathcal{D}_{1:t-1})$ is inferred at the previous timestep $t - 1$. The estimate of a changepoint conditioned on the past changes and the measurements is

$$
\begin{aligned}
&p(s_t = 1 \,|\, s_{1:t-1}, \mathcal{D}_{1:t}) \\
&\quad = \frac{p(s_t = 1)p(\boldsymbol{y}_t \,|\, \boldsymbol{x}_t, s_{1:t-1}, s_t = 1, \mathcal{D}_{1:t-1})}{p(s_t = 1)p(\boldsymbol{y}_t \,|\, s_t = 1, \boldsymbol{x}_t, s_{1:t-1}, \boldsymbol{y}_{1:t-1}) + p(s_t = 0)p(\boldsymbol{y}_t \,|\, s_t = 0, \boldsymbol{x}_t, s_{1:t-1}, \mathcal{D}_{1:t-1})} \\
&\quad = \left(1 + \exp\left(-\log\left(\frac{p(s_t = 1)p(\boldsymbol{y}_t \,|\, s_t = 1, \boldsymbol{x}_t, s_{1:t-1}, \boldsymbol{y}_{1:t-1})}{p(s_t = 0)p(\boldsymbol{y}_t \,|\, s_t = 0, \boldsymbol{x}_t, s_{1:t-1}, \mathcal{D}_{1:t-1})}\right)\right)\right)^{-1} = \sigma(m_t),
\end{aligned}
\tag{61}
$$

where $\sigma(x) = 1/(1 + \exp(-x))$ and

$$m_t = \log\left(\frac{p(\boldsymbol{y}_t \mid s_t = 1, \boldsymbol{x}_t, s_{1:t-1}, \mathcal{D}_{1:t-1})}{p(\boldsymbol{y}_t \mid s_t = 0, \boldsymbol{x}_t, s_{1:t-1}, \mathcal{D}_{1:t-1})}\right) + \log\left(\frac{p(s_t = 1)}{p(s_t = 0)}\right), \tag{62}$$

and similarly,

$$p(s_t = 0 \mid s_{1:t-1}, \mathcal{D}_{1:t}) = 1 - \sigma(m_t). \tag{63}$$

Finally, the transition between states is given by $p(s_{1:t} \mid s_{1:t-1}) = p(s_t)$.

## B  Algorithms

---

**Algorithm 2** Implementation of `RL[K]-PR`. We consider an update at time $t$ and one-step ahead forecasting at time $t + 1$ under a Gaussian linear model with known observation variance.

---

**Require:** $(\boldsymbol{\mu}_0, \boldsymbol{\Sigma}_0)$ // default prior beliefs
**Require:** $\mathcal{D}_t = (\boldsymbol{x}_t, \boldsymbol{y}_t)$ // current observation
**Require:** $\{r_{t-1}^{(k)}\}_{k=1}^K \in \{0, \ldots, t-1\}^K$ // bank of runlengths at time $t - 1$
**Require:** $\{p(r_{t-1}^{(k)}, \mathcal{D}_{1:t})\}_{k=1}^K$ // joint from past hypotheses
**Require:** $\left\{(\boldsymbol{\mu}_{t-1}^{(k)}, \boldsymbol{\Sigma}_{t-1}^{(k)})\right\}_{k=1}^K$ // beliefs from past hypotheses
**Require:** $\boldsymbol{x}_{t+1}$ // next-step observation
**Require:** $p(\boldsymbol{y} \mid \boldsymbol{\theta}, \boldsymbol{x}) = \mathcal{N}(\boldsymbol{y} \mid \boldsymbol{\theta}^\intercal \boldsymbol{x}, \mathbf{R}_t)$ // Choice of (M.1: model)
1: // Evaluate hypotheses if there is no changepoint
2: **for** $k = 1, \ldots, K$ **do**
3:   $r_t^{(k)} \leftarrow r_{t-1}^{(k)} + 1$
4:   $p(\boldsymbol{y}_t \mid r_t^{(k)}, \boldsymbol{x}_t, \mathcal{D}_{1:t-1}) \leftarrow \mathcal{N}(\boldsymbol{y}_t \mid \boldsymbol{x}_t^\intercal \boldsymbol{\mu}_{t-1}^{(k)}, \boldsymbol{x}_t^\intercal \boldsymbol{\Sigma}_{t-1}^{(k)} \boldsymbol{x}_t + \mathbf{R}_t)$ // posterior predictive for $k$-th hypothesis
5:   $p(r_t^{(k)}, \mathcal{D}_{1:t}) \leftarrow p(\boldsymbol{y}_t \mid r_t^{(k)}, \boldsymbol{x}_t, \mathcal{D}_{1:t-1}) \, p(r_{t-1}^{(k)}, \mathcal{D}_{1:t-1}) \, p(r_t^{(k)} \mid r_{t-1}^{(k)})$ // update joint density
6:   $(\bar{\boldsymbol{\mu}}_t^{(k)}, \bar{\boldsymbol{\Sigma}}_t^{(k)}) \leftarrow (\boldsymbol{\mu}_{t-1}^{(k)}, \boldsymbol{\Sigma}_{t-1}^{(k)})$
7:   $\tau_t(\boldsymbol{\theta}_t; r_t^{(k)}) \leftarrow \mathcal{N}(\boldsymbol{\theta}_t \mid \bar{\boldsymbol{\mu}}_t, \bar{\boldsymbol{\Sigma}}_t)$ // choice of (M.3: prior)
8:   $q_t(\boldsymbol{\theta}_t; r_t^{(k)}) \propto \tau_t(\boldsymbol{\theta}_t; r_t^{(k)}) \, p(\boldsymbol{y}_t \mid \boldsymbol{\theta}^\intercal \boldsymbol{x}_t, \mathbf{R}_t) \propto \mathcal{N}(\boldsymbol{\theta}_t \mid \boldsymbol{\mu}_t^{(k)}, \boldsymbol{\Sigma}_t^{(k)})$ // following (26)
9: **end for**
10: // Evaluate hypothesis under a changepoint
11: $r_t^{(k+1)} \leftarrow 0$
12: $p(\boldsymbol{y}_t \mid r_t^{(k+1)}, \boldsymbol{x}_t, \mathcal{D}_{1:t-1}) \leftarrow \mathcal{N}(\boldsymbol{y}_t \mid \boldsymbol{x}_t^\intercal \boldsymbol{\mu}_0, \boldsymbol{x}_t^\intercal \boldsymbol{\Sigma}_0 \boldsymbol{x}_t + \mathbf{R}_t)$ // posterior predictive for $k$-th hypothesis
13: $p(r_t^{(k+1)}, \mathcal{D}_{1:t}) \leftarrow p(\boldsymbol{y}_t \mid r_t^{(k+1)}, \boldsymbol{x}_t, \mathcal{D}_{1:t-1}) \sum_{k=1}^K p(r_t^{(k)}, \mathcal{D}_{1:t}) \, p(r_t^{(t+1)} \mid r_t^{(k)} - 1)$
14: // Extend number of hypotheses to $K + 1$ and keep top $K$ hypotheses
15: $I_{1:k} = \text{top.k}(\{p(r_t^{(1)}, \mathcal{D}_{1:t}), \ldots, p(r_t^{(k+1)}, \mathcal{D}_{1:t})\}, K)$
16: $\{p(r_t^{(k)}, \mathcal{D}_{1:t})\}_{k=1}^K \leftarrow \text{slice.at}(\{p(r_t^{(k)}, \mathcal{D}_{1:t})\}_{k=1}^{K+1}, I_{1:K})$
17: $\{(\boldsymbol{\mu}_t^{(k)}, \boldsymbol{\Sigma}_t^{(k)})\}_{k=1}^K \leftarrow \text{slice.at}(\{(\boldsymbol{\mu}_t^{(k)}, \boldsymbol{\Sigma}_t^{(k)})\}_{k=1}^{K+1}, I_{1:K})$
18: // build weight and make prequential prediction
19: $\nu_t(r_t^{(k)}) \leftarrow \frac{p(r_t^{(k)}, \mathcal{D}_{1:t})}{\sum_{j=1}^K p(r_t^{(j)}, \mathcal{D}_{1:t})}$ for $k = 1, \ldots, K$
20: $\hat{\boldsymbol{y}}_{t+1} \leftarrow \boldsymbol{x}_{t+1}^\intercal \left(\sum_{k=1}^K \nu_t(r_t^{(k)}) \boldsymbol{\mu}_t^{(k)}\right)$ // prequential prediction under a linear-Gaussian model
21: **return** $\{(\boldsymbol{\mu}_t^{(k)}, \boldsymbol{\Sigma}_t^{(k)}, r_t^{(k)})\}_{k=1}^K, \hat{\boldsymbol{y}}_{t+1}$

---

In Algorithm 2, the function top.k$(A, K)$ returns the indices of the top $K \geq 1$ elements of $A$ with highest value. The function slice.at$(A, B)$ returns the elements in $A$ according to the list of indices $B$. If $|A| \leq |B|$, we return all elements in $A$.

---

**Algorithm 3** Implementation of `RL[K]-MMPR`. We consider an update at time $t$ and one-step ahead forecasting at time $t+1$ under a Gaussian linear model with known observation variance.

---

**Require:** $\mathcal{D}_t = (\boldsymbol{x}_t, \boldsymbol{y}_t)$ // current observation

**Require:** $\{r_{t-1}^{(k)}\}_{k=1}^K \in \{0, \ldots, t-1\}^K$ // bank of runlengths at time $t-1$

**Require:** $\{p(r_{t-1}^{(k)}, \mathcal{D}_{1:t})\}_{k=1}^K$ // joint from past hypotheses

**Require:** $\left\{(\boldsymbol{\mu}_{t-1}^{(k)}, \boldsymbol{\Sigma}_{t-1}^{(k)})\right\}_{k=1}^K$ // beliefs from past hypotheses

**Require:** $\boldsymbol{x}_{t+1}$ // next-step observation

**Require:** $p(\boldsymbol{y} \,|\, \boldsymbol{\theta}, \boldsymbol{x}) = \mathcal{N}(\boldsymbol{y} \,|\, \boldsymbol{\theta}^\intercal \boldsymbol{x}, \mathbf{R}_t)$ // Choice of (M.1: model)

1: // Evaluate hypotheses if there is no changepoint
2: **for** $k = 1, \ldots, K$ **do**
3:      $r_t^{(k)} \leftarrow r_{t-1}^{(k)} + 1$
4:      $p(\boldsymbol{y}_t \,|\, r_t^{(k)}, \boldsymbol{x}_t, \mathcal{D}_{1:t-1}) \leftarrow \mathcal{N}(\boldsymbol{y}_t \,|\, \boldsymbol{x}_t^\intercal \boldsymbol{\mu}_{t-1}^{(k)}, \boldsymbol{x}_t^\intercal \boldsymbol{\Sigma}_{t-1}^{(k)} \boldsymbol{x}_t + \mathbf{R}_t)$ // posterior predictive for $k$-th hypothesis
5:      $p(r_t^{(k)}, \mathcal{D}_{1:t}) \leftarrow p(\boldsymbol{y}_t \,|\, r_t^{(k)}, \boldsymbol{x}_t, \mathcal{D}_{1:t-1}) \, p(r_{t-1}^{(k)}, \mathcal{D}_{1:t-1}) \, p(r_t^{(k)} \,|\, r_{t-1}^{(k)})$ // update joint density
6:      $(\bar{\boldsymbol{\mu}}_t^{(k)}, \bar{\boldsymbol{\Sigma}}_t^{(k)}) \leftarrow (\boldsymbol{\mu}_{t-1}^{(k)}, \boldsymbol{\Sigma}_{t-1}^{(k)})$
7:      $\tau_t(\boldsymbol{\theta}_t; r_t^{(k)}) \leftarrow \mathcal{N}(\boldsymbol{\theta}_t \,|\, \bar{\boldsymbol{\mu}}_t, \bar{\boldsymbol{\Sigma}}_t)$ // choice of (M.3: prior)
8:      $q_t(\boldsymbol{\theta}_t; r_t^{(k)}) \propto \tau_t(\boldsymbol{\theta}_t; r_t^{(k)}) \, p(\boldsymbol{y}_t \,|\, \boldsymbol{\theta}^\intercal \boldsymbol{x}_t, \mathbf{R}_t) \propto \mathcal{N}(\boldsymbol{\theta}_t \,|\, \boldsymbol{\mu}_t^{(k)}, \boldsymbol{\Sigma}_t^{(k)})$ // following (26)
9: **end for**
10: // Evaluate hypothesis under a changepoint
11: $r_t^{(k+1)} \leftarrow 0$
12: $\boldsymbol{\mu}_0 \leftarrow \mathbb{E}[\boldsymbol{\theta}_t \,|\, r_t, \boldsymbol{y}_{1:t-1}]$ // following (56)
13: $\boldsymbol{\Sigma}_0 \leftarrow \mathbb{E}[\boldsymbol{\theta}_t \boldsymbol{\theta}_t^\intercal \,|\, r_t, \boldsymbol{y}_{1:t-1}] - (\mathbb{E}[\boldsymbol{\theta}_t \,|\, r_t, \boldsymbol{y}_{1:t-1}])(\mathbb{E}[\boldsymbol{\theta}_t \,|\, r_t, \boldsymbol{y}_{1:t-1}])^\intercal$ // following (56) and (57)
14: $p(\boldsymbol{y}_t \,|\, r_t^{(k+1)}, \boldsymbol{x}_t, \mathcal{D}_{1:t-1}) \leftarrow \mathcal{N}(\boldsymbol{y}_t \,|\, \boldsymbol{x}_t^\intercal \boldsymbol{\mu}_0, \boldsymbol{x}_t^\intercal \boldsymbol{\Sigma}_0 \boldsymbol{x}_t + \mathbf{R}_t)$ // posterior predictive for $k$-th hypothesis
15: $p(r_t^{(k+1)}, \mathcal{D}_{1:t}) \leftarrow p(\boldsymbol{y}_t \,|\, r_t^{(k+1)}, \boldsymbol{x}_t, \mathcal{D}_{1:t-1}) \sum_{k=1}^K p(r_t^{(k)}, \mathcal{D}_{1:t}) \, p(r_t^{(t+1)} \,|\, r_t^{(k)} - 1)$
16: // Extend number of hypotheses to $K+1$ and keep top $K$ hypotheses
17: $I_{1:k} = \text{top.k}(\{p(r_t^{(1)}, \mathcal{D}_{1:t}), \ldots, p(r_t^{(k+1)}, \mathcal{D}_{1:t})\}, K)$
18: $\{p(r_t^{(k)}, \mathcal{D}_{1:t})\}_{k=1}^K \leftarrow \text{slice.at}(\{p(r_t^{(k)}, \mathcal{D}_{1:t})\}_{k=1}^{K+1}, I_{1:K})$
19: $\{(\boldsymbol{\mu}_t^{(k)}, \boldsymbol{\Sigma}_t^{(k)})\}_{k=1}^K \leftarrow \text{slice.at}(\{(\boldsymbol{\mu}_t^{(k)}, \boldsymbol{\Sigma}_t^{(k)})\}_{k=1}^{K+1}, I_{1:K})$
20: // build weight and make prequential prediction
21: $\nu_t(r_t^{(k)}) \leftarrow \dfrac{p(r_t^{(k)}, \mathcal{D}_{1:t})}{\sum_{j=1}^K p(r_t^{(j)}, \mathcal{D}_{1:t})}$ for $k = 1, \ldots, K$
22: $\hat{\boldsymbol{y}}_{t+1} \leftarrow \boldsymbol{x}_{t+1}^\intercal \left( \sum_{k=1}^K \nu_t(r_t^{(k)}) \boldsymbol{\mu}_t^{(k)} \right)$ // prequential prediction under a linear-Gaussian model
23: **return** $\{(\boldsymbol{\mu}_t^{(k)}, \boldsymbol{\Sigma}_t^{(k)}, r_t^{(k)})\}_{k=1}^K, \hat{\boldsymbol{y}}_{t+1}$

---

---

**Algorithm 4** Implementation of `RL[1]-OUPR*`, with update at time $t$ and for one-step ahead forecasting at time $t + 1$, under a Gaussian linear model with known observation variance.

---

**Require:** $\mathcal{D}_t = (\boldsymbol{x}_t, \boldsymbol{y}_t)$ // current observation
**Require:** $\boldsymbol{x}_{t+1}$ // next-step observation
**Require:** $\epsilon \in (0, 1)$ // restart threshold
**Require:** $r_{t-1} \in \{0, \ldots, t - 1\}$ // runlength at time $t - 1$
**Require:** $(\boldsymbol{\mu}_0, \boldsymbol{\Sigma}_0)$ // default prior beliefs
**Require:** $(\boldsymbol{\mu}_{t-1}, \boldsymbol{\Sigma}_{t-1})$ // beliefs from prior step
**Require:** $p(\boldsymbol{y} \,|\, \boldsymbol{\theta}, \boldsymbol{x}) = \mathcal{N}(\boldsymbol{y} \,|\, \boldsymbol{\theta}^\intercal \boldsymbol{x}, \mathbf{R}_t)$ // Choice of (M.1: model)
1: $(r_t^{(0)}, r_t^{(1)}) \leftarrow (0, r_{t-1} + 1)$ // choice of (M.2: auxvar)
2: $p(\boldsymbol{y}_t \,|\, r_t^{(0)}, \boldsymbol{x}_t, \mathcal{D}_{1:t-1}) \leftarrow \mathcal{N}(\boldsymbol{y}_t \,|\, \boldsymbol{x}_t^\intercal \boldsymbol{\mu}_0, \boldsymbol{x}_t^\intercal \boldsymbol{\Sigma}_0 \, \boldsymbol{x}_t + \mathbf{R}_t)$ // posterior predictive at changepoint
3: $p(\boldsymbol{y}_t \,|\, r_t^{(1)}, \boldsymbol{x}_t, \mathcal{D}_{1:t-1}) \leftarrow \mathcal{N}(\boldsymbol{y}_t \,|\, \boldsymbol{x}_t^\intercal \boldsymbol{\mu}_{t-1}, \boldsymbol{x}_t^\intercal \boldsymbol{\Sigma}_{t-1} \, \boldsymbol{x}_t + \mathbf{R}_t)$ // posterior predictive if no changepoint
4: $\nu_t(r^{(1)}) \leftarrow \dfrac{p(\boldsymbol{y}_t \,|\, r_t^{(1)}, \boldsymbol{x}_t, \mathcal{D}_{1:t-1})(1-\pi)}{p(\boldsymbol{y}_t \,|\, r_t^{(1)}, \boldsymbol{x}_t, \mathcal{D}_{1:t-1})\,(1-\pi) + p(\boldsymbol{y}_t \,|\, r_t^{(0)}, \boldsymbol{x}_t, \mathcal{D}_{1:t-1})\,\pi}$ // probability of no-changepoint at timestep $t$
5:
6: **if** $\nu(r_t^{(1)}) > \epsilon$ **then**
7: $\quad r_t \leftarrow r_t^{(1)}$
8: $\quad \bar{\boldsymbol{\mu}}_t^{(r_t)} \leftarrow \boldsymbol{\mu}_{t-1}^{(r_{t-1})} \, \nu(r_t^{(1)}) + \boldsymbol{\mu}_0 \left(1 - \nu(r_t^{(1)})\right)$
9: $\quad \bar{\boldsymbol{\Sigma}}_t^{(r_t)} \leftarrow \boldsymbol{\Sigma}_{t-1}^{(r_{t-1})} \, \nu(r_t^{(1)})^2 + \boldsymbol{\Sigma}_0 \left(1 - \nu(r_t^{(1)})^2\right)$
10: **else if** $\nu(r_t^{(1)}) \leq \epsilon$ **then**
11: $\quad r_t \leftarrow r_t^{(0)}$
12: $\quad \bar{\boldsymbol{\mu}}_t^{(r_t)} \leftarrow \boldsymbol{\mu}_0$
13: $\quad \bar{\boldsymbol{\Sigma}}_t^{(r_t)} \leftarrow \boldsymbol{\Sigma}_0$
14: **end if**
15: $\tau_t(\boldsymbol{\theta}_t; r_t) \leftarrow \mathcal{N}(\boldsymbol{\theta}_t \,|\, \bar{\boldsymbol{\mu}}_t, \bar{\boldsymbol{\Sigma}}_t)$ // choice of (M.3: prior)
16: $q_t(\boldsymbol{\theta}_t; r_t) \propto \mathcal{N}(\boldsymbol{\theta}_t \,|\, \bar{\boldsymbol{\mu}}_t, \bar{\boldsymbol{\Sigma}}_t)\, p(\boldsymbol{y}_t \,|\, \boldsymbol{\theta}^\intercal \boldsymbol{x}_t, \mathbf{R}_t) \propto \mathcal{N}(\boldsymbol{\theta}_t \,|\, \boldsymbol{\mu}_t, \boldsymbol{\Sigma}_t)$ // choice of (A.1: posterior)— via (26)
17: $\hat{\boldsymbol{y}}_{t+1} \leftarrow \boldsymbol{x}_{t+1}^\intercal \boldsymbol{\mu}_t$ // prequential prediction (given linear-Gaussian model)
18: **return** $(\boldsymbol{\mu}_t, \boldsymbol{\Sigma}_t, r_t), \hat{\boldsymbol{y}}_{t+1}$

---

## C  Bayesian inference when the dynamics are known

If the dynamics of model parameters $p(\boldsymbol{\theta}_{t+1}, \,|\, \boldsymbol{\theta}_t, \psi_{t+1})$ and the dynamics for auxiliary variable $p(\psi_{t+1} \,|\, \psi_t)$ are known, then a full-Bayesian treatment of the prediction at time $t + 1$ is given by *predicting* the expected model parameters and then performing inference.

Specifically, given the prior belief state $p(\psi_t, \boldsymbol{\theta}_t \,|\, \mathcal{D}_{1:t})$ and the new input $\boldsymbol{x}_{t+1}$, the predictive distribution over the output $\boldsymbol{y}_{t+1}$ is given by

$$p(\boldsymbol{y}_{t+1} \,|\, \boldsymbol{x}_{t+1}, \mathcal{D}_{1:t}) = \sum_{\psi_{t+1}} \int_{\boldsymbol{\theta}_{t+1}} p(\boldsymbol{y}_{t+1} \,|\, \boldsymbol{x}_{t+1}, \boldsymbol{\theta}_{t+1}) p(\psi_{t+1}, \boldsymbol{\theta}_{t+1} \,|\, \mathcal{D}_{1:t}) \, \mathrm{d}\boldsymbol{\theta}_{t+1}, \tag{64}$$

where the parameter predictive distribution is

$$p(\psi_{t+1}, \boldsymbol{\theta}_{t+1} \,|\, \mathcal{D}_{1:t}) = \sum_{\psi_t} \int_{\boldsymbol{\theta}_t} p(\psi_{t+1}, \boldsymbol{\theta}_{t+1} \,|\, \psi_t, \boldsymbol{\theta}_t) p(\psi_t, \boldsymbol{\theta}_t \,|\, \mathcal{D}_{1:t}) \, \mathrm{d}\boldsymbol{\theta}_t. \tag{65}$$

Here, the parameter dynamics factorises as

$$p(\psi_{t+1}, \boldsymbol{\theta}_{t+1} \,|\, \psi_t, \boldsymbol{\theta}_t) = p(\psi_{t+1} \,|\, \psi_t) p(\boldsymbol{\theta}_{t+1} \,|\, \boldsymbol{\theta}_t, \psi_{t+1}), \tag{66}$$

and the prior belief state factorises as

$$p(\psi_t, \boldsymbol{\theta}_t \,|\, \mathcal{D}_{1:t}) = p(\psi_t \,|\, \mathcal{D}_{1:t}) p(\boldsymbol{\theta}_t \,|\, \psi_t, \mathcal{D}_{1:t}). \tag{67}$$

Combining all the above equations, we can derive an expression for the posterior predicted mean

$$
\begin{aligned}
\hat{\boldsymbol{y}}_{t+1} &= \mathbb{E}[\boldsymbol{y}_{t+1} \mid \boldsymbol{x}_{t+1}, \mathcal{D}_{1:t}] \\
&= \sum_{\psi_{t+1}} \sum_{\psi_t} p(\psi_{t+1} \mid \psi_t) p(\psi_t \mid \mathcal{D}_{1:t}) \\
&\quad \times \int_{\boldsymbol{\theta}_{t+1}} \int_{\boldsymbol{\theta}_t} h(\boldsymbol{\theta}_{t+1}, \boldsymbol{x}_{t+1}) p(\boldsymbol{\theta}_{t+1} \mid \boldsymbol{\theta}_t, \psi_{t+1}) p(\boldsymbol{\theta}_t \mid \psi_t, \mathcal{D}_{1:t}) \, \mathrm{d}\boldsymbol{\theta}_t \, \mathrm{d}\boldsymbol{\theta}_{t+1}.
\end{aligned}
\tag{68}
$$

After making the prediction, we observe the outcome $\boldsymbol{y}_{t+1}$, and then update the belief state for the parameter $\boldsymbol{\theta}_{t+1}$ (for each value of $\boldsymbol{\psi}_{t+1}$) using

$$
p(\boldsymbol{\theta}_{t+1} \mid \psi_{t+1}, \mathcal{D}_{1:t+1}) = \frac{1}{Z_{t+1}^{\psi_{t+1}}} p(\boldsymbol{y}_{t+1} \mid \boldsymbol{x}_{t+1}, \boldsymbol{\theta}_{t+1}) p(\boldsymbol{\theta}_{t+1} \mid \psi_{t+1}, \mathcal{D}_{1:t}),
$$

$$
Z_{t+1}^{\psi_{t+1}} = p(\boldsymbol{y}_{t+1} \mid \boldsymbol{x}_{t+1}, \mathcal{D}_{1:t}, \psi_{t+1}) = \int_{\boldsymbol{\theta}_{t+1}} p(\boldsymbol{y}_{t+1} \mid \boldsymbol{x}_{t+1}, \boldsymbol{\theta}_{t+1}) p(\boldsymbol{\theta}_{t+1} \mid \psi_{t+1}, \mathcal{D}_{1:t}) \, \mathrm{d}\boldsymbol{\theta}_{t+1}, \tag{69}
$$

$$
p(\boldsymbol{\theta}_{t+1} \mid \psi_{t+1}, \mathcal{D}_{1:t}) = \int_{\boldsymbol{\theta}_t} p(\boldsymbol{\theta}_{t+1} \mid \boldsymbol{\theta}_t, \psi_{t+1}) p(\boldsymbol{\theta}_t \mid \mathcal{D}_{1:t}) \, \mathrm{d}\boldsymbol{\theta}_t.
$$

Similarly we update the belief state for the auxiliary variable using

$$
p(\psi_{t+1} \mid \mathcal{D}_{1:t+1}) = \frac{1}{Z_{t+1}} p(\boldsymbol{y}_{t+1} \mid \boldsymbol{x}_{t+1}, \psi_{t+1}) p(\psi_{t+1} \mid \mathcal{D}_{1:t}),
$$

$$
Z_{t+1} = p(\boldsymbol{y}_{t+1} \mid \boldsymbol{x}_{t+1}, \mathcal{D}_{1:t}) = \sum_{\psi_{t+1}} Z_{t+1}^{\psi_{t+1}} p(\psi_{t+1} \mid \mathcal{D}_{1:t}), \tag{70}
$$

$$
p(\psi_{t+1} \mid \mathcal{D}_{1:t}) = \sum_{\psi_t} p(\psi_{t+1} \mid \psi_t) \, p(\psi_t \mid \mathcal{D}_{1:t}).
$$

This full Bayesian treatment is different from BONE, because we do not assume well-specified dynamics. Thus, BONE makes predictions at time $t+1$ with only the posterior at time $t$, i.e., it does not require a *predict* step.

