# OpenReview forum: "A unifying framework for generalised Bayesian online learning in non-stationary environments"
_TMLR — Accepted by TMLR_

### Review · Reviewer_j55f · 2025-01-06

**Summary Of Contributions:**

This paper introduces BONE (Bayesian Online learning in Non-stationary Environments), a unifying framework that integrates diverse methodologies for addressing non-stationarity in Bayesian learning. The contributions are:
Unified software and model framework: BONE presents a modular structure comprising three modeling components (measurement model, auxiliary process, and conditional prior) and two algorithmic components (posterior estimation and auxiliary variable weighting). This provides a comprehensive view of existing approaches under a common framework.

The authors propose novel approach (which should be highlighted among the dense literature) RL[1]-OUPR*, a method designed to handle mixed non-stationary data regimes by combining mechanisms for gradual and abrupt changes. Then theydemonstrated the adaptability of BONE across multiple tasks, including segmentation and forecasting, with RL[1]-OUPR* achieving state-of-the-art performance.

**Audience:**

Yes

**Broader Impact Concerns:**

No.

**Claims And Evidence:**

Yes

**Requested Changes:**

1)Scalability in Large-Scale Settings: Although theoretical generality is demonstrated, there is little discussion on practical efficiency or scalability for high-dimensional data or long-horizon tasks. In section 2.8 the authors talked about several fitting methods, without discussing the possbile scalability issue (which is the best and why), which is a key question in change-point detection. It would be helpful to discuss the computational complexity of posterior estimation and auxiliary weighting in high-dimensional settings. For instance, how does BONE handle scaling issues in DA[inf] models where hypothesis spaces grow exponentially?

2) Strengthen the introduction by explicitly stating how BONE diverges from changepoint-specific methods like Bayesian online changepoint detection (e.g., Adams & MacKay 2007). In addition, there are lines from other communities treating similar problems, if the authors can make in-depth, more focused review, perhaps focus on change-poing literature.
From the SciML community, recent efforts have been extended to auxilliary-free change-point in online setting:
Luo et.al (2024) Non-smooth Bayesian optimization in tuning scientific applications, The International Journal of High Performance Computing Applications. From the statistical community, Hartigan (1993) A bayesian analysis for change point problems. Journal of the American Statistical Association. (which is more appropriate than Barry&Hartigan 1993 cited now.)
The lack of in-depth literature review in the introduction section makes readers confused the problem setting: how the non-stationarity BONE attempts to tackle, is different from non-smooth/change-point data generating process. Non-stationary stochastic process can be continuouts and even smooth.

3)On the other hand, Section 2.3 is un-necessarily lengthy, but still does not make it clear how it is different from existing change-point or online literature. The use of the term "continual learning" is argubly too broad, since the OCL usually means that the underlying data generating mechanism could shift. For another example, it does not make a compelling case for how auxiliary variables (e.g., RL) differ from existing changepoint models. I would suggest to condense this section by focusing on the essential differences and remove overly broad discussions of continual learning, which dilute the focus. It may be more clear if the comparison Table 2 is shown at the beginning, then short explanations about the difference follow after.

4) From a practical perspective, it would be good to provide heuristics or examples for choosing auxiliary processes like RL[1] or CPP for different types of non-stationarity. I think this is a  key component since it represents the evolution of non-stationary latent states. Practitioners will benefit from clear guidelines on tailoring these components to specific applications. Or the author could give several examples where some well-known methods cannot handle (e.g., GP with non-stationary kernels when underlying environment has shifts) and then the BONE handles well.

5) Error Analysis for Mixed Regimes: It would be interesting to see a qualitative breakdown of errors when handling both gradual and abrupt non-stationarity in the same dataset. This would better demonstrate RL[1]-OUPR*'s robustness. Since there lacks usual theory (I understood that for a highly complicated model, theoretic guarantee may not be feasible), there seems more room for refined error analysis.
6) How does the current BONE scales when the input lives in higher dimensions?

Points 1 to 4 must be addressed.

**Strengths And Weaknesses:**

Strengths:
1)The BONE framework unifies diverse methods, bridging gaps between changepoint detection, continual learning, and adaptive forecasting, and making it accessible to both ML and statistical communities.
2)The RL[1]-OUPR* algorithm innovatively reconciles gradual and abrupt non-stationarity, addressing limitations in existing approaches that focus on only one type of change. The experimental section spans various real-world tasks, offering robust evidence of the framework's versatility. They also claimed to provide a library implemented in Jax enhances accessibility and reproducibility for researchers and practitioners.

Weaknesses:
1) The paper references changepoint methods (e.g., Adams & MacKay, Gupta et al.) but does not clearly articulate how BONE innovates upon these approaches.
2) Non-stationary processes addressed by BONE appear conflated with changepoint detection, leading to ambiguity in the problem setting. while the introduction highlights the modularity of BONE, it lacks a focused literature review to clarify how the framework differs from prior work in Bayesian online learning (e.g., dynamic priors, ensemble Kalman filters).
3)Terms like "continual learning" are used broadly but are not directly tied to the problem of non-stationarity. This may confuse readers unfamiliar with OCL and its overlap with BONE.

---

> ### Author Response · Authors · 2025-02-17
> **Response to reviewer**
>
> We would like to thank the reviewer for their constructive feedback and for highlighting areas that could benefit from clarification.
>
> A recurring theme in the review is the overlap between changepoint detection methods and our framework. For instance, the reviewer remarked that "Non-stationary processes addressed by BONE appear conflated with changepoint detection.” In the new version we are more explicit when explaining that many methods developed for changepoint detection—as well as the other approaches outlined in Section 2.3, such as filtering, and non-stationary bandits—can be subsumed into the BONE framework. This enables methods from various subfields of machine learning to be applied to a broader range of sequential online learning problems (across domains) where non-stationarity may be present.
>
> To ensure this point is clear, we have revised the introduction and abstract to provide greater clarity on this key idea.
>
> Below, we respond to points (1-6) raised by the reviewer.
>
> ## R1.1
> Here, we address the reviewer’s comments regarding scalability and high-dimensional settings.
>
> Firstly, with respect to the choice of (A.1), in the revised version of the manuscript, we will reference Table 3 in [4] that shows various options for (A.1) and their associated computational complexity. This should provide a clearer understanding of the trade-offs and practical efficiency of different choices.
>
> Secondly, regarding the scalability of DA[inf] models (where hypothesis spaces grow exponentially), we discuss one potential approach on Page 11 (Section 2.9.1) in the paragraph titled “Low-memory variants — from DA[inf] to DA[K].” This section outlines how low-memory variants can address these scaling challenges. We will also cite sample-based methods such as the Rao-Blackwellized particle filter [5].
>
> As for long-horizon tasks, our approach focuses on sequential updates, where algorithms process the next step (and subsequent steps) while updating weights and auxiliary variables as needed. Under this formulation, the algorithms are designed to run indefinitely, without a predefined time horizon.
>
> ## R1.2
> **R1.2.1 Strengthening the introduction to clarify how BONE diverges from changepoint-specific methods (e.g., BOCD)**
> Thank you for this remark. BONE is a general framework designed to address a wide range of sequential online learning problems. Changepoint detection methods, such as Bayesian online changepoint detection (BOCD), can be formulated as specific instances of BONE (e.g., RL-PR in our experiments, as shown in Table 2).
>
> However, unlike changepoint-detection methods, which often focus on identifying the location of abrupt transitions in data, BONE provides a unifying framework that accommodates diverse types of changes in the data. For example, while RL-PR tackles abrupt changes, other BONE methods, such as C-ACI, are designed to address gradual or continuous transitions assuming a constant change, which is never directly detected, but rather assumed as a modelling choice.
>
> To ensure these distinctions are clear, we revised the introduction to highlight how BONE generalises beyond changepoint-specific methods like BOCD and provides a broader conceptual foundation for tackling sequential learning problems.
>
>
> **R1.2.2 Literature review and specific references (Luo et al., 2024; Hartigan, 1993)**
> We appreciate the reviewer’s suggestion to include Luo et al. (2024) and Hartigan (1993) in our discussion. Regarding Luo et al. (2024), their work focuses on global optimisation, whereas BONE is designed for sequential prediction requiring data indexed by time. Some BONE methods, such as C-ACI, gradually forget past observations, making global fitting infeasible. For this reason, Bayesian optimisation methods like those in Luo et al. (2024) fall outside the scope of our framework.
>
> In the new version, we cite their work and clarify this distinction.
>
> **R1.2.3 Clarifying the non-stationarity addressed by BONE and the problem setting**
> BONE is a flexible framework that does not impose a specific type of non-stationarity on the data. As stated in Section 2.3, the choice of components (e.g., M.2 and M.3) allows users to adapt the framework to tackle a variety of scenarios, including abrupt changes (e.g., RL-PR), gradual or smooth transitions (e.g., C-ACI), and gradual and abrupt (e.g., OUPR). In this sense, BONE goes beyond changepoint-specific methods by providing a toolkit that can address a broader range of non-stationary processes.
>
> To ensure this is clear to readers, we revised the introduction to state how BONE accommodates both abrupt and continuous forms of non-stationarity. Additionally, we clarified the scope of the problem and highlighted the distinction between BONE’s sequential prediction focus and other approaches (such as global optimisation methods).

---

> > ### Author Response · Authors · 2025-02-17
> > **Response to reviewer**
> >
> > ## R1.3
> > **“On the other hand, Section 2.3 is un-necessarily lengthy, but still does not make it clear how it is different from existing change-point or online literature**
> > Thank you for your remark. We would like to clarify that the purpose of BONE is to unify these literatures under a common framework. Section 2.3 is central to this goal, as it explains how methods developed for various sequential tasks—changepoint detection, filtering, and non-stationary bandits, among others—can all be expressed as specific instances of BONE.
> >
> > In the new version, we reworded parts of Section 2 to streamline the discussion and we removed some unnecessary material.
> >
> > **“it does not make a compelling case for how auxiliary variables (e.g., RL) differ from existing changepoint models”**
> > The goal of BONE is not to propose auxiliary variables that differ from those used in changepoint detection models. Instead, BONE aims to generalise and accommodate auxiliary-variable-based methods, including those commonly used in changepoint detection. For example, as illustrated in Table 2, auxiliary variables like RL (runlength)  are a natural fit within the BONE framework, where they can be interpreted and applied across a variety of sequential problems, not just changepoint detection. Other auxiliary variables we include are CPP (changepoint probability) and CPL (changepoint location).
> >
> > In the revised manuscript, we introduced a new diagram capturing the information in Table 2 to highlight how auxiliary variables like RL are generalised within BONE and how this supports its unifying nature. We believe that this should clarify the relationship between auxiliary-variable-based changepoint models and the broader framework of BONE.
> >
> > **“The use of the term "continual learning" is argubly too broad, since the OCL usually means that the underlying data generating mechanism could shift [...]. I would suggest to condense this section by focusing on the essential differences and remove overly broad discussions of continual learning, which dilute the focus.”**
> > Thank you for the observation. We included the remarks on the breadth of OCL made by the reviewer. In the new version we also streamlined the discussion in Section 2.
> >
> > **It may be more clear if the comparison Table 2 is shown at the beginning, then short explanations about the difference follow after**
> > Thank you for the remark. In the version we included an easy-to-digest diagram that motivates what comes later in Table 2. The new diagram is in the introduction.
> >
> > ## R1.4
> > **It would be good to provide heuristics or examples for choosing auxiliary processes**
> > We thank the reviewer for this suggestion. In the revised manuscript, we provided heuristics for selecting auxiliary processes (M.2) based on prior knowledge of the type of non-stationarity.
> >
> > ## R1.5
> > **Error Analysis for Mixed Regimes**
> > The electricity forecasting dataset in Section 4.1.1 provides a "real-world" example with an abrupt changepoint (the start of the COVID-19 pandemic) followed by gradual reversion to normal electricity load levels.
> >
> > We will add a table summarising the errors made before Covid (no changepoint), during Covid (abrupt changepoint), and after Covid (gradual regime change).
> >
> > ## R1.6
> > **“How does the current BONE scales when the input lives in higher dimensions?”**
> > The scalability of BONE in higher dimensions depends directly on the choice of (A.1). BONE itself is not inherently affected by changes in dimensionality (either in parameter space or input space), but specific choices within the framework may be.
> >
> > In the new version of the paper we included a reference to Table 3 in [4] that provides a detailed scalability analysis of different choices for (A.1) (see also R1.1).

---

### Review · Reviewer_jXLu · 2025-02-08

**Summary Of Contributions:**

This paper is mostly a review extracting a common structure from articles about *Bayesian online learning in non-stationary environments*. The paper distinguishes 3 different elements:
 1. (M1) the model performing the prediction (given an input), whose parameters are denoted by $\theta$;
 2. (M2) the "auxiliary variables" $\psi \in \Psi$, containing the information about the non-stationarity;
 3. (M3) the "conditional prior" over $\theta$, which depends on $\psi$ and information on the past.

Also, the paper distinguishes 2 methods to choose to compute:
 1. (A1) the posterior distribution of $\theta$, given the conditional prior and an input data point;
 2. (A2) the weights $\nu(\psi)$ over the different auxiliary variables $\psi \in \Psi$.

The paper reviews the various models and methods solving at least one of these 5 elements of the problem.

Additionally, the paper proposes a new model for M3, named RL[1]-OUPR*.

The Experiments section focuses on evaluation of RL[1]-OUPR*, mainly in a problem of electricity forecasting, and additionally on synthetic datasets.

**Audience:**

Yes

**Claims And Evidence:**

Yes

**Requested Changes:**

1. Is there any existing review in the same area?
 2. Is there any Bayesian online algorithm that does not fit into BONE?
 3. Explain the idea of the algorithms when it is lacking.
 4. Explain the selection of algorithms made in Section 4.
 5. Discuss the purpose of the separation (M1, M2, M3, A1, A2) (classification of existing algorithms? designing new algos? ...)

**Strengths And Weaknesses:**

# Strengths

## Clarity

Overall, the paper is well structured and easy to understand.

## Contribution and relevance

This paper reviews many different Bayesian methods that can be used in online learning, which could fit into two kinds of TMLR papers:
 1. development of new analytical frameworks that advance theoretical studies of practical learning methods;
 2. surveys that draw new connections, highlight trends, and suggest new problems in an area.

However, although this paper proposes a new analytical framework, there is no clear development towards further theoretical study. So, it should probably be considered as a survey.

Anyway, the review work is useful and the "BONE" formalism makes it easier to classify the various techniques.

Also, RL[1]-OUPR* is competitive with the other methods in the tested settings.

## Originality

This work is original.

## Soundness

The experimental section shows that RL[1]-OUPR* is competitive with the other methods in several settings, including a non-synthetic one.

# Weaknesses

## Clarity

p. 6 and after: formal definitions of $\boldsymbol{\mu}$ and $\boldsymbol{\Sigma}$ should be introduced somewhere.

## Contributions

Since this paper reviews existing techniques in Bayesian online learning, it should cite relevant reviews that already exist, if any (if not, the absence of such reviews should be indicated in the Introduction).

The possible limitations of the proposed formalism BONE are not evoked. For instance, is there any Bayesian online learning technique that does not fit BONE?

The justification (either theoretical or experimental) of the non-trivial methods is not always present. Examples: C-ACI, ME-LSSM.

The Experiments Section is not entirely adapted to a review paper such as this one. For instance, the selection of the algorithms to be tested is not explained or motivated. Are the selected algorithms known for performing better than other?

## Soundness

The elements (M1, M2, M3) of the BONE framework are not bricks that can be stacked independently: there are some combinations of "auxiliary variables" M2 and "conditional priors" M3 that are incompatible (either because they are useless or because they are meaningless). For instance, ME-Static is theoretically possible, but is (intuitively) useless to do, and C-PR does not make any sense.
So, presenting an algorithm as combination (M1, M2, M3, A1, A2) is potentially misleading. The same remarks holds for the choice of the algorithm A1.

It is true that the choice of (M1, M2, M3, A1, A2) can be regarded as a way to classify existing algorithms, but it cannot really be used to build new algorithms from existing bricks. So, it is unsure that the BONE *formalism* would be helpful for future research (which is distinct from the *review* itself).

## Minor errors

p. 2: "Our estimate of the mean" -> "Our estimate of the expectation"

Section 3: "tackle any of the the problems mention in Section 2.3" -> "tackle any of the problems mentioned in Section 2.3"

Section 2.8.3: "Bayesian online natural gradien" -> "Bayesian online natural gradient"

---

> ### Author Response · Authors · 2025-02-17
> **Response to reviewer**
>
> We would like to thank the reviewer for their constructive feedback and for highlighting areas that could benefit from clarification.
>
> Below we collect our responses.
>
> ## R2.1
> **Is there any existing review in the same area?**
> As far as we know, there are no existing reviews in the area of Bayesian online learning. There are, however, many reviews on how to tackle “non-stationarity” in different subfields of statistics and machine learning.
>
> In the revised version of the paper we cited relevant existing reviews from these other subfields of statistics and machine learning in the section “Example tasks that can be solved with BONE”.
>
> ## R2.2
> **Is there any Bayesian online algorithm that does not fit into BONE.**
> In the new version of the paper we added a graphical model that explains the interaction of the various objects within BONE. Bayesian online algorithms with an underlying graphical model that cannot be written in the form we propose, would not fall into the BONE framework. Examples of these are, for instance [6],  [7], or [8] (provided by Reviwer j55f).
>
> We will make this point clearer in the new version of the paper.
>
> ## R2.3
> **Explain the idea of the algorithms when it is lacking.**
> For a description of some of the methods we use, please see the table “List of methods we compare in our experiments”.
> In the final version, we will expand on the idea behind the algorithms we use where required.
>
> ## R2.4
> **Explain the selection of algorithms made in Section 4.**
> For most of the experiments we selected classical methods (RL-PR), simple baselines (C-ACI), and modern algorithms (CPP-OU) to compete against ours (RL[1]-SPR).
>
> In the new version of the paper we made this point clearer.
>
> ## R2.5
> **Discuss the purpose of the separation (M1, M2, M3, A1, A2)**
> The purpose of separation is twofold: classification of existing algorithms and designing new algorithms.
> For example, Table 2 presents many algorithms as instances of M1, M2, A1, and A2.
>
> In the new version, we clarified the importance of this modularity in the abstract and introduction.
>
> ## R2.6
> **Soundness**
> Regarding the soundness of some methods, while it is true that some combinations may be “useless”, this does not mean that the choices of components are incompatible. It simply means that some choices are strongly preferable to others depending on the situation. We will make a note on this in the new version.
>
> ## R2.7
> **Other comments and minor errors**
> Thank you for spotting these. In the new version introduce mu, Sigma and correct the typos.

---

### Review · Reviewer_dPSS · 2025-02-11

**Summary Of Contributions:**

This interesting paper proposes a unifying online learning framework that applies to both stationary and non-stationary environments.  It is particularly concerned with applying Bayesian inference to non-stationary regression problems where the non-stationarity is caused by a non-stationary discrete latent variable and conditional on the parameters and the discrete latent variable a (stationary) regression model is applied.

The paper then does a comprehensive review of the literature and shows that many existing algorithms can be formulated within their ("BONE") framework.  They also show that the framework allows the development of new algorithms and demonstrate them on both an electricity dataset and a bandit problem.

**Audience:**

Yes

**Claims And Evidence:**

Yes

**Requested Changes:**

This is my (provisional) attempt to write down the core insight of BONE.  My request is to ask if this line of reasoning can be used to improve the clarity of the paper.  I may not be completely correct here, but I think amendments to the paper presenting the approximate inference in these terms could be very enlightening.

I will try to rewrite the model only using the index to implement non-stationary.  Starting with the non-stationary case we have:
$p(y_{1:t},\theta|x_1:t) = p(\theta) \prod_i^t p(y_i|x_i,\theta)$

The posterior is:
$P(\theta|x_{1:t},y_{1:t}) \propto p(\theta) \prod_i^t p(y_i|x_i,\theta)$
when we incorporate non-stationary, it becomes the case that $\theta$ can change in time (so we index it $theta_t$), but the new parameter $\psi$ does not change in time.
$p(y_{1:t},\theta_{1:t},\psi|x_1:t) = p(\psi) \prod_i^t p(y_i|x_i,\theta_i)p(\theta_i|\psi,x_i)$
The posterior is then
$P(\theta_{1:t},\psi|x_{1:t},y_{1:t}) \propto p(\psi) \prod_i^t p(y_i|x_i,\theta_i)p(\theta_i|\psi,x_i)$

Importantly $\theta$ is now indexed, but $\psi$ is not.

This paper is concerned mostly with recursive updates (which is one reason why product signs never appear in the paper – also the paper uses the word likelihood for what I think is more traditionally called the model)
The update of $\theta_t,\psi$ has a recursive form which exploits the fact that we can summarize past knowledge with:
$p(\psi|y_{1:t-1},x_{1:t-1})$, and we can safely neglect the posterior on $\theta_{1:t-1}$.

Given $x_t$ a prediction of $y_t$ can be made with:
$\int \int p(y_t|x_t,\theta_t)p(\theta_t|\psi,x_t) p(\psi|y_{1:t-1},x_{1:t-1}) d \theta_t d\psi$

We can also update knowledge of $\psi$ with
$p(\psi,\theta_t|y_{1:t},x_{1:t}) \propto p(y_t|x_t,\theta_t) p(\theta_t|\psi,x_t) p(\psi|y_{1:t-1},x_{1:t-1})$, which we can marginalize to $p(\psi|y_{1:t},x_{1:t})$ to enable the next update.

BONE applies in the simple but useful case where $\psi$ is discrete with $l$ states.  This means that there are $l$ mixture components  $p(\theta_t|\psi=1,x_t),...,p(\theta_t|\psi=l,x_t)$.  BONE propose an approximate inference algorithm framework for solving this problem (Algorithm 1), it then performs a thorough literature review and shows many methods are instances of BONE (Table 2) and proposes and tests some new algorithm/models that belong to the BONE framework.

**Strengths And Weaknesses:**

This paper is well written in the sense of writing style, structure and the mathematical notation is consistent and the paper is well structured.  The authors appear to have a good command of the literature from an algorithmic point of view and make numerous insightful observations comparing an impressive number of different algorithms and models.

The experiments appear to be well conducted and the examples are interesting.

What I find to be the main limitation of the paper is that the paper is not motivated in a particularly clear way from a usual Bayesian standpoint (despite the title of the paper).  I recognize that footnote 1 is intended to explain this, although I must confess I find this footnote confused as to what "Bayesian" means.  As the main contribution of this paper is to provide a unifying framework I do find this limitation concerning, but I do think it is fixable with (significant) revisions.  I should add here, that my taste and view is not always in concordance with others so If other referees or the editor disagree with me here I will happily defer to them.  In any event, I will elaborate on my concerns below.

A second point, the author(s) appear to suggest that the BONE framework not only allows cataloguing methods as per Table 2, but also building new methods like LEGO.  Is this the case?  If it is, then it seems like a very useful contribution (reminiscent of the graphical model work popular a decade ago) that should be developed a little more strongly.  Perhaps the JAX library (redacted for now) implements exactly this, this doesn't come through strongly enough in the paper!

Given this important contribution, I think it is worth considering a presentation that is easier to digest from the Bayesian point of view, which would mean addressing some of these oddities:
* I don't understand the use of the $h(\theta_t;x_t)$ notation (Equation 5).  It seems to hide the model and tie the use to squared loss / expectation.  While it is used in Equation 22-26, it does seem to me like a more traditional approach would be both clearer and more general.
* If I am not wrong, parameters are indexed for two reasons.  One is to allow non-stationarity i.e. theta is changing over time so it is indexed theta_t; and a second reason, knowledge of theta is changing over time.  I assume this is the reason that the indexing is present even in the stationary case (Equation 1 and 2).  I think it would prefer the clarity significantly to only use the indexing by t for the first situation and use conditioning to illustrate the information used to inform the distribution of the parameter.  For this reason I think theta ought to be indexed but and not psi
* Algorithm 1 seems to reduce predicting y to a finite mixture and computes the weights and parameters to the mixing components.  This would be much easier to follow if it was taken back to first principles, Starting with the joint posterior on $\theta$ and $\psi$ and then introducing approximations would likely help with the explanation.  I attempt to outline how this might look in the next section, but please correct or clarify..
* The paper in many places uses the word "likelihood" where I think the author(s) mean "model".
* Actually writing down the likelihood might help clarify the paper (with a product of i.i.d densities conditional on a fixed in the stationary case and a product sign of i.i.d densities conditioned on a moving parameter in other cases).  I understand the paper focuses on recursive updates, but this can still help explain the approximate inference being applied.
* The term "conditional prior M3" in section 2.7 is potentially confusing (although the contradictory nature of the name should give the reader a hint to pay attention).  If I understand correctly the author(s) mean the choice of parametric form used for approximate recursive posterior updates.  I think the fundamental insight of BONE is that under certain assumptions exact or approximate recursive updates are possible. A bit more care here needed could greatly improve clarity.  The term conditional prior is confusing as it mixes the concept of approximate inference and an approximate parametric posterior (say using VB) and actual model assumptions in the usual sense of a prior.
* The non-stationarity in the paper AFAICT applies to the case of stationarity if the "regime" is known, but is non-stationary when it is unknown, this is an interesting case but there are others.  Please correct me if I am wrong here.
* Table 2 is quite important to defend the power of the unifying framework yet it is not easy to read.

Equation 7 does not say what the expectation is under.
Equation 8 shouldn’t $\theta$ be a $d$ by $q$ matrix (why the transpose on theta in equation 8?).

---

> ### Author Response · Authors · 2025-02-17
> **Response to reviewer**
>
> We thank the reviewer for their thoughtful and constructive feedback. In particular, we appreciate the pointers to make our framework easier to digest from a Bayesian point of view.
>
> We answer the reviewer’s questions below.
>
> ## R3.1
> **“What I find to be the main limitation of the paper is that the paper is not motivated in a particularly clear way from a usual Bayesian standpoint [...] I think amendments to the paper presenting the approximate inference in these terms could be very enlightening”**
> We agree with the idea of motivating our method from a more “usual Bayesian standpoint”. To this end, and as requested by the reviewer, we made a significant revision of motivating ideas behind BONE in Section 2. In particular, we now introduce a probabilistic graphical model that shows how both the model parameters (theta(t)) and the auxiliary variables (psi(t)) are indexed by time.
>
> Next, in line with the reviewer’s suggestion, we wrote down the joint posterior and derived the Bayesian expected posterior predictive. Finally, we introduced additional modelling choices that are possible to make in BONE and how they deviate from the classical Bayesian perspective.
>
> ## R3.2
> **“BONE framework not only allows cataloguing methods as per Table 2, but also building new methods like LEGO. Is this the case?”**
> This is exactly the case. We appreciate the reviewers comment on making this point stand out and we will be more forthcoming in the revised version.
>
> ## R3.3
> **I don't understand the use of the h(theta; xt) notation (Equation 5)**
> In the new version, we added clarity and stated that h(theta, x) is parametrised by theta and that we use it to model the conditional mean for y given x.
>
> ## R3.4
> **“If I am not wrong, parameters are indexed for two reasons. One is to allow non-stationarity [...]; and a second reason, knowledge of theta is changing over time. [...] I think theta ought to be indexed but and not psi”**
> In our framework, parameters (theta) are indexed to reflect recursive updates (akin to traditional filtering, where the time index denotes the search for the best local–in time–set of parameters), and auxiliary variables (psi) are indexed to track how to adapt to non-stationarity, e.g., keeping the time since the last changepoint. A good example of an auxiliary variable whose options increase (exponentially) with time is CPL.
>
> ## R3.5
> **Algorithm 1 seems to reduce predicting y to a finite mixture and computes the weights and parameters to the mixing components. This would be much easier to follow if it was taken back to first principles.**
> We have re-written Section 2 from a first-principles approach. Please see R3.1 above.
>
> ## R3.6
> **The paper in many places uses the word "likelihood" where I think the author(s) mean "model".**
> Thank you for raising this point, we have modified the text to mention (M.1) as the choice of “model”.
>
> ## R3.7
> **Actually writing down the likelihood might help clarify the paper**
> We have done this in Section 2. Please R3.1 above
>
> ## R3.8
> **The term "conditional prior M3" in section 2.7 is potentially confusing [...] as it mixes the concept of approximate inference and an approximate parametric posterior (say using VB) and actual model assumptions in the usual sense of a prior”**
> We agree that this point requires greater clarity. In BONE, the conditional prior approximates what is referred to, in the Bayesian filtering literature, as the posterior predictive. It serves as an inductive bias on the state-space model (SSM), governing the evolution of parameters after time t-1 but before an observation is made at time t. Essentially, it acts as the prior that is combined with the likelihood of the data at time t. However, rather than solving the integration step explicitly, many works in the literature make an ad hoc modelling choice for the evolution of parameters conditioned on an auxiliary variable, often assuming a Gaussian form. This is why we refer to it as a modelling choice rather than an algorithmic one.
>
> We will clarify this distinction in the section where we introduce conditional priors.
>
> ## R3.9
> **The non-stationarity in the paper AFAICT applies to the case of stationarity if the "regime" is known, but is non-stationary when it is unknown, this is an interesting case but there are others. Please correct me if I am wrong here**
> This is correct. Thus, we capture non-stationarity by considering multiple hypotheses at once with multiple values of the auxiliary variable psi_t, weighing each hypothesis through the choice of (A.2).
>
> ## R3.10
> **Table 2 is quite important [...] yet is not easy to read.**
> Thank you for this remark. In the new version we added a diagram that highlights the type of non-stationarity that we have found in the literature. In particular, the diagram shows whether the auxiliary variable is continuous or discrete and then a high-level description of what “non-stationarity” is being modelled, along with references in the literature.

---

### Author Response · Authors · 2025-02-17
**General comment**

We thank all three reviewers for their constructive feedback on our paper. We were happy to see that all three reviewers answered “yes” to the statements that (i) the “claims made in the submission were supported by accurate, convincing and clear evidence” and that (ii) “individuals in TMLR's audience would be interested in the findings of this paper.”

The reviewers highlighted the following strengths of BONE:
* “This paper is well written in the sense of writing style, structure and the mathematical notation is consistent and the paper is well structured. The authors appear to have a good command of the literature from an algorithmic point of view and make numerous insightful observations comparing an impressive number of different algorithms and models.”  — Reviewer dPSS
* “The BONE framework unifies diverse methods, bridging gaps between changepoint detection, continual learning, and adaptive forecasting, and making it accessible to both ML and statistical communities” — Reviewer j55f
* “The BONE framework not only allows cataloguing methods as per Table 2, but also building new methods [...] it seems like a very useful contribution” — Reviewer dPSS
* “The RL[1]-OUPR* algorithm innovatively reconciles gradual and abrupt non-stationarity, addressing limitations in existing approaches that focus on only one type of change” — Reviewer j55f
* “The review work is useful and the ‘BONE’ formalism makes it easier to classify the various techniques” — Reviewer jXLu
* “RL[1]-OUPR* is competitive with the other methods in the tested settings”. — Reviewer jXLu

The reviewers also requested a number of changes to the paper. For example:
* BONE should be derived from a “classical Bayesian perspective” — Reviewer dPSS
* Highlight Table 2 and make it more digestible— Reviewer j55f and Reviewer dPSS
* Highlight the strength in BONE in unifying various methods throughout the different literature and the ability of BONE to enable modular development of new methods — Reviewer jXLu, Reviewer dPSS, and Reviewer j55f
* Highlight that the Jax library will be released after the review process  — Reviewer dPSS

All these requested changes have been addressed in the new version.

In the revised version, we will highlight that BONE is designed as a general framework and a conceptual approach to model-building. This perspective also allows us to introduce novel algorithms, such as RL[1]-OUPR*. Following the generalised Bayes literature [1][2][3], we have updated the title to “A unifying view on generalised Bayesian online learning [...]”, which reinforces the justification for relaxing the classical Bayesian assumptions and still allowing us to incorporate methods utilising aspects form the Bayesian machinery.

In the new version, we believe that we addressed all of your requested changes. In each of our responses below, we provide details about each of the actions taken with respect to each of the requested changes.
We also uploaded the current new version of the article with the new material in red (for the convenience of the referees). Although we will be polishing this material, we thought it would be good that you see our preliminary changes.

We hope you find this new revision suitable for publication in TMLR.

Please consider the following references for some of our responses
* [1]: Knoblauch, Jeremias, Jack Jewson, and Theodoros Damoulas. "An optimization-centric view on Bayes' rule: Reviewing and generalizing variational inference." Journal of Machine Learning Research 23.132 (2022): 1-109.
* [2]: Khan, Mohammad Emtiyaz, and Håvard Rue. "The Bayesian learning rule." Journal of Machine Learning Research 24.281 (2023): 1-46.
* [3] Bissiri, Pier Giovanni, Chris C. Holmes, and Stephen G. Walker. "A general framework for updating belief distributions." Journal of the Royal Statistical Society: Series B (Statistical Methodology) 78.5 (2016): 1103-1130.
* [4] Jones, Matt, Peter Chang, and Kevin Murphy. "Bayesian Online Natural Gradient (BONG)." arXiv preprint arXiv:2405.19681 (2024).
* [5] Murphy, Kevin, and Stuart Russell. "Rao-Blackwellised particle filtering for dynamic Bayesian networks." Sequential Monte Carlo methods in practice. New York, NY: Springer New York, 2001. 499-515.
* [6] Vilmarest, Joseph de, and Olivier Wintenberger. "Viking: variational Bayesian variance tracking." Statistical Inference for Stochastic Processes (2024): 1-22.
* [7] Scalzo, Bruno, et al. "Nonstationary portfolios: Diversification in the spectral domain." ICASSP 2021-2021 IEEE International Conference on Acoustics, Speech and Signal Processing (ICASSP). IEEE, 2021.
* [8] Luo, Hengrui, et al. "Non-smooth bayesian optimization in tuning problems." arXiv preprint arXiv:2109.07563 (2021).

---

> ### Comment · Reviewer_dPSS · 2025-02-18
> **Thanks it's much clearer**
>
> I now understand much better the use of $h()$.
>
> A few comments relevant to Equation 8 and the paragraph above.
>
> The likelihood cannot always be computed as a discrepancy between $h()$ and the observed $y$ depending on the model.
>
> Just an issue of taste.  The "classic Bayesian" approach sharply distinguishes modelling and decision making i.e. models, decision rules and losses, where "generalized Bayes" does not (the negative loss is instead used as a quasi-log likelihood as in Equation 7).  As well as loosing the separation between decision making and inference, there are other strange properties here ..  multiplying the loss by 2 will square the posterior etc ..

---

> > ### Author Response · Authors · 2025-02-18
> > **Thank you, we agree.**
> >
> > We completely agree that “the likelihood cannot always be computed as discrepancy between h() and the observed  data”. In fact, we write  “that ℓ(yt; θt, xt) can be a loss function that measures the discrepancy […]“.
> >
> > We will make this point clearer along the lines of your remark.

---

### Decision · Action_Editor_d933 · 2025-03-10

**Recommendation:** Accept as is

**Comment:**

There has been some debate between the reviewers and with the authors regarding the nature of the contribution. In light of the interesting discussions that took place, I think the proposed manuscript revision makes it clearer that it does not correspond to a review or extensive literature survey (in particular the text does not discuss theoretical performance guarantees for any of the methods). On the other hand, it does propose an original view (with companion implementation library), aimed at unifying the many algorithms proposed in the literature for recursive prediction under assumed non-stationary behavior. Again, an interesting topic that has been discussed is the "Bayesianism" of the approach and this is also a point that has been clarified: the general point of view is Bayesian (as most often in filtering and prediction) but the emphasis is put on scalable recursive computations (typically not through Monte Carlo methods) and there is an interesting taxonomy (which is part of the proposed BONE framework) of various approximate prediction updates that can be performed, depending on the nature of the underlying non-stationarity representation. While it does not provide, strictly speaking, new methods the proposed BONE framework is modular and can be used to assemble together ideas that were proposed in different contexts (as is done in the experimentation section).

Overall, this is a valuable contribution to the field of applied online prediction in non-stationary environments that deserves tor be published in TMLR.

**Audience:**

The paper will be of interest to readers interested in practical online (ie. not retrospective) prediction approaches in non-stationary environments.

**Claims And Evidence:**

The paper is clearly written with precise formalism and claims.